# Environmental fluctuations reshape an unexpected diversity-disturbance relationship in a microbial community

Christopher P Mancuso[1], Hyunseok Lee[2], Clare I Abreu[2], Jeff Gore[2], Ahmad S Khalil[1,3]*

[1]Department of Biomedical Engineering and Biological Design Center, Boston University, Boston, United States; [2]Department of Physics, Massachusetts Institute of Technology, Cambridge, United States; [3]Wyss Institute for Biologically Inspired Engineering, Harvard University, Boston, United States

**Abstract** Environmental disturbances have long been theorized to play a significant role in shaping the diversity and composition of ecosystems. However, an inability to specify the characteristics of a disturbance experimentally has produced an inconsistent picture of diversity-disturbance relationships (DDRs). Here, using a high-throughput programmable culture system, we subjected a soil-derived bacterial community to dilution disturbance profiles with different intensities (mean dilution rates), applied either constantly or with fluctuations of different frequencies. We observed an unexpected U-shaped relationship between community diversity and disturbance intensity in the absence of fluctuations. Adding fluctuations increased community diversity and erased the U-shape. All our results are well-captured by a Monod consumer resource model, which also explains how U-shaped DDRs emerge via a novel 'niche flip' mechanism. Broadly, our combined experimental and modeling framework demonstrates how distinct features of an environmental disturbance can interact in complex ways to govern ecosystem assembly and offers strategies for reshaping the composition of microbiomes.

**\*For correspondence:**
khalil@bu.edu

## Introduction

Biodiversity is a cornerstone of ecosystem stability and function (*Tilman et al., 2014*). While it is well appreciated that environmental changes influence species diversity in all ecosystems, the exact nature of this critical relationship is unclear. Without a predictive understanding of how ecosystems respond to perturbations, we are poorly prepared for environmental changes of anthropogenic origin, such as rising global temperatures (*Hoegh-Guldberg and Bruno, 2010*), and unable to design effective and robust interventions in ecosystems, such as microbiomes of medical or agricultural importance (*Lemon et al., 2012*; *Widder et al., 2016*). Accordingly, there have been many efforts aimed at understanding the role of environmental disturbances, which are perturbations to the state of an environment. These disturbances are of ecological interest for the impact they have on a community, for example, by bringing about mortality of organisms and a reduction of biomass of a community. Various diversity-disturbance (DDR) relationships have been proposed that draw intuition from observations of natural ecosystems. DDRs describe how community diversity depends on disturbance intensity, which we define as the mean mortality rate imposed by the disturbance over time. A famous example is the Intermediate Disturbance Hypothesis (*Connell, 1978*; *Huston, 1979*), in which species diversity peaks at intermediate disturbance intensities (*Figure 1a*). However, DDRs derived from observational studies of disparate ecosystems and disturbance regimes often have inconsistent results (*Mackey and Currie, 2001*; *Hughes et al., 2007*). Earlier assertions that disturbance weakens or interrupts competition (*Connell, 1978*; *Huston, 1979*) have been refuted by both

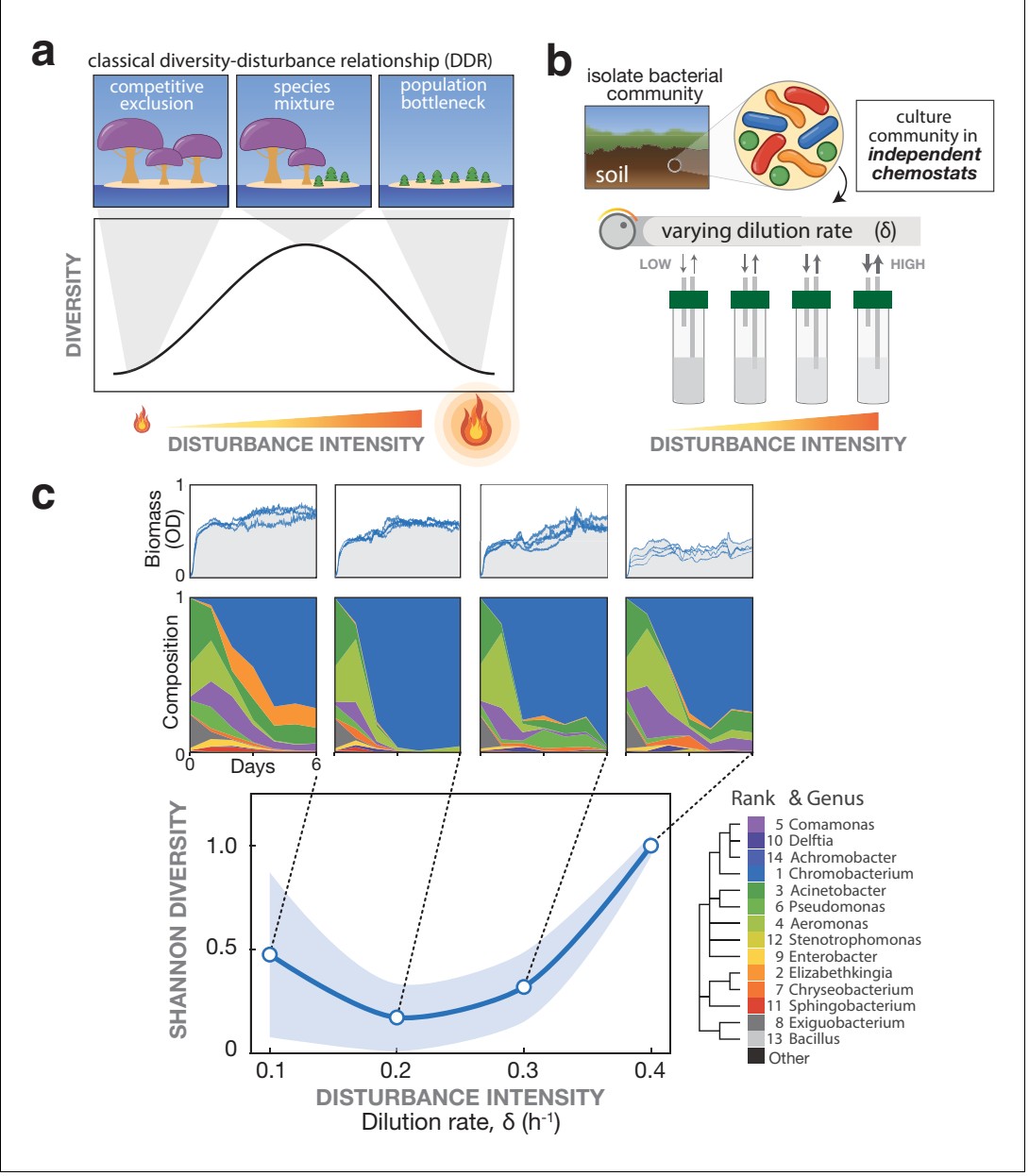

**Figure 1.** Emergence of a U-shaped diversity-disturbance relationship (DDR) in a microbial community for constantly applied disturbance at different intensities. (a) Different DDRs have been proposed based on observations of natural ecosystems, including the Intermediate Disturbance Hypothesis in which diversity peaks at intermediate disturbance levels, see depiction. (b) In the laboratory, microbial communities can be cultivated and subjected to varying disturbance intensity levels by tuning the dilution rate in chemostats. (c) A bacterial community exhibits a U-shaped diversity dependence on the disturbance intensity. Samples of a soil-derived bacterial community were grown for 6 days in eVOLVER mini-chemostats at four different dilution rates. Top: Optical density over time quantifies biomass for replicate cultures. Middle: Mean relative abundance of bacterial genera from replicate cultures, determined by 16S sequencing. Mean rank abundance is denoted by order, taxonomic similarity is denoted by color. Bottom: Plotting the endpoint number of species (Amplicon Sequence Variants) vs. dilution rate produces a U-shaped curve, rather than a peaked DDR. Shaded window indicates a one standard deviation confidence interval.

theory (*Chesson and Huntly, 1997*; *Fox, 2013*) and empirical findings (*Violle et al., 2010*) that harsher environments instead reinforce dependence on limiting factors. Without a quantitative framework that directly pairs theory and experiment, it has been difficult to determine the source of disagreement between the many conflicting predictions and observations surrounding DDRs.

Importantly, the impact of a disturbance on an ecosystem depends on the disturbance characteristics. For example, although some environmental disturbances do not vary over time, many environmental disturbances occur with lower frequencies and introduce fluctuations that drive an ecosystem in and out of different states. The environmental fluctuations associated with a disturbance may in fact stabilize communities by creating temporal niches, similar to seasonal effects (*Levins, 1979*; *Chesson, 1994*). Indeed, coexistence can be promoted in a fluctuation-dependent manner due to storage effects (e.g. dormancy in poor conditions) or if species exhibit relative non-linearities in their competitive responses (e.g. differently shaped growth curves) (*Chesson, 1994*; *Letten et al., 2018*). Yet, coexistence might also arise from the overall time-averaged disturbance intensity in a fluctuation-independent manner (*Chesson and Huntly, 1997*; *Fox, 2013*). To determine whether the effects of disturbance on diversity are truly fluctuation-dependent (*Chesson, 2000*), a disturbance should ideally be decomposed into distinct components of *mean intensity* (e.g. time-averaged disturbance magnitude) and *frequency* (e.g. temporal profile of fluctuations). Indeed, experimental findings (*Benedetti-Cecchi et al., 2006*; *Hall et al., 2012*) and theory (*Miller et al., 2011*) have suggested that diverse DDRs could arise when considering these factors independently. There is therefore a need for comprehensive, controlled studies which pair theory with experimental methods to produce datasets that can deconvolve the effects of intensity and fluctuation.

Laboratory experiments offer a greater degree of control and throughput compared to field studies, particularly for tractable ecosystem models like microbial communities (*De Roy et al., 2014*). Microbes are easily quantified with next-generation sequencing (*Gohl et al., 2016*; *Bolyen et al., 2019*; *Callahan et al., 2016*; *Janssen et al., 2018*), and have been widely used in the laboratory to model community assembly (*Friedman et al., 2017*; *Venturelli et al., 2018*; *Goldford et al., 2018*), cross-feeding relationships (*Mee et al., 2014*), and succession (*Chuang et al., 2019*). Laboratory models have also linked changes in diversity in response to fluctuating nutrient levels (*Sommer, 1985*; *Grover, 1988*) and disturbances such as sonication (*Violle et al., 2010*), ultraviolet radiation (*Gibbons et al., 2016*), osmotic pressures (*Letten et al., 2018*), or toxic compounds (*Santillan et al., 2019*). Dilution is perhaps the most common choice for a laboratory disturbance, as it causes species-independent mortality and replenishes the system with fresh nutrients, reminiscent of flow in soil, aquatic, or gut microbiomes. Unlike disturbances with indirect biological impacts (such as pH, temperature, or osmolarity disturbances), there is a direct link between the dilution disturbance event (removal of culture volume) and the biological outcome (mortality of community members). In simple batch culture experiments, where cultures remain undisturbed except for a periodic dilution step, coexistence has been observed at intermediate dilution levels (*Gibbons et al., 2016*; *Abreu et al., 2019*), although different DDRs arise under different dilution regimes (*Hall et al., 2012*), suggesting that the dilution parameter space is vastly under-sampled. For more precise tuning of dilution or other parameters, experimentalists have long relied on continuous culture methods (*Sommer, 1985*; *Grover, 1988*; *Monod, 1950*); unfortunately, these systems have traditionally been intractable to large-scale, multidimensional experiments. Recently, we developed eVOLVER, a flexible and automated continuous culture platform that enables independent control over conditions in a large number of mini-bioreactors (*Wong et al., 2018*; *Heins et al., 2019*), thus opening up the possibility to explore microbial community dynamics under controlled, multidimensional environmental disturbances. By programming different dilution profiles with eVOLVER, we set out to independently quantify the effects of disturbance intensity (i.e. average dilution rate) and frequency (i.e. temporal profile of dilution rate) on the composition and diversity of microbial communities.

## Results

### Measuring microbial community diversity in constantly applied disturbances reveals a U-shaped relationship

First, we sought to measure microbial diversity at various mean disturbance intensities in the absence of fluctuations by culturing a community under different mean dilution rates at the same near-continuous frequency. We cultivated replicate samples of a soil-derived microbiome in separate eVOLVER bioreactor arrays in dilute Nutrient Broth for six days (comprising 20–90 generations), during which continuously diluted cultures approached a stable composition (*Figure 3—figure supplements 1* and *2*). In a chemostat, the flow of media into the vessel is matched by flow of spent media and cells out of the vessel, so disturbance intensity is directly related to dilution rate (*Figure 1b*). We thus varied the disturbance intensity by varying the dilution rate across the arrays (see Materials and methods, *Figure 3—figure supplements 1* and *2*). We sampled cultures daily and used 16S sequencing to quantify composition and diversity over time. As expected, we observed decreasing biomass of the cultures at increasing dilution rates (*Figure 1c* and *Figure 3—figure supplement 3*). Surprisingly, after quantifying the composition of each culture, we observed a U-shaped diversity-disturbance relationship (*Figure 1c*), with the number of surviving species at intermediate dilution rate at roughly half of the number at either low or high dilution rate. We were particularly intrigued because a U-shaped relationship between diversity and disturbance intensity does not agree with historical wisdom (*Huston, 1979*), although U-shaped DDRs have been reported for other disturbance characteristics such as frequency (*Miller et al., 2011*). Although U-shaped DDRs are uncommon in empirical observations (*Mackey and Currie, 2001*; *Hughes et al., 2007*), the conditions under which we observed it were quite straightforward: constantly applied disturbance. Thus, to better understand our observation, we sought a modeling framework in which a U-shaped DDR could emerge from constantly applied disturbance, while still capturing other reported DDR shapes.

### A consumer-resource model captures experimentally observed DDRs and uncovers a niche-flip mechanism for conditional coexistence

To develop a theoretical framework for understanding and predicting DDR behavior, we simulated microbial co-cultures in bioreactor conditions. Before examining higher order systems, we started by examining the simplest case that could give rise to a U-shaped DDR: a two-species competition where coexistence breaks down at intermediate disturbance levels. This simple system illustrates a sufficient condition that leads to the U-shaped DDR, and extensive analysis on a broader set of conditions is provided in Appendix 1 and 2. To link changes in disturbance intensity to changes in competitive outcomes, we turned to consumer resource models (*Tilman, 1982*; *MacArthur, 1970*). In consumer resource models, species growth rates are a function of resource concentrations. Resource depletion rates are in turn a function of species abundances, so species interact by competing for the same limiting resources. The range of resource concentrations that can support growth of a species can be graphically analyzed on a Tilman diagram (*Tilman, 1982*) by defining a Zero Net Growth Isocline (ZNGI) (*Figure 2a*). Briefly, the ZNGIs intersect with each axis in the Tilman diagram at c* values, which are the concentration of a particular resource necessary for a species to survive at the specified mortality rate. Similarly, the ZNGIs define the boundary of supplied resource combinations that will support growth of a species. As a population consumes resources, the resource concentrations move from the resource supply point toward the ZNGI, where growth rate is equal to the mortality rate (i.e. disturbance intensity) and the population is at equilibrium. The shape and location of these ZNGIs also predict the outcome of competition, as resource consumption by the population can cause equilibrium resource levels to cross the ZNGI of one species, leading to exclusion by the other (*Figure 2a*). If the ZNGIs intersect with each other, coexistence is possible. The borders of the coexistence region are invasion boundaries where either species can invade the other; in this region, consumption brings equilibrium resource concentrations towards the intersection of the ZNGIs (*Figure 2a*).

Disturbance affects the ZNGIs in multiple ways. First, since higher mortality rates require higher growth rates to maintain equilibrium, more resources are required to maintain equilibrium. Accordingly, the ZNGIs shift toward higher resource levels at higher disturbance intensities. Second, for saturating growth models, the growth rate of each species may change relative to each other as

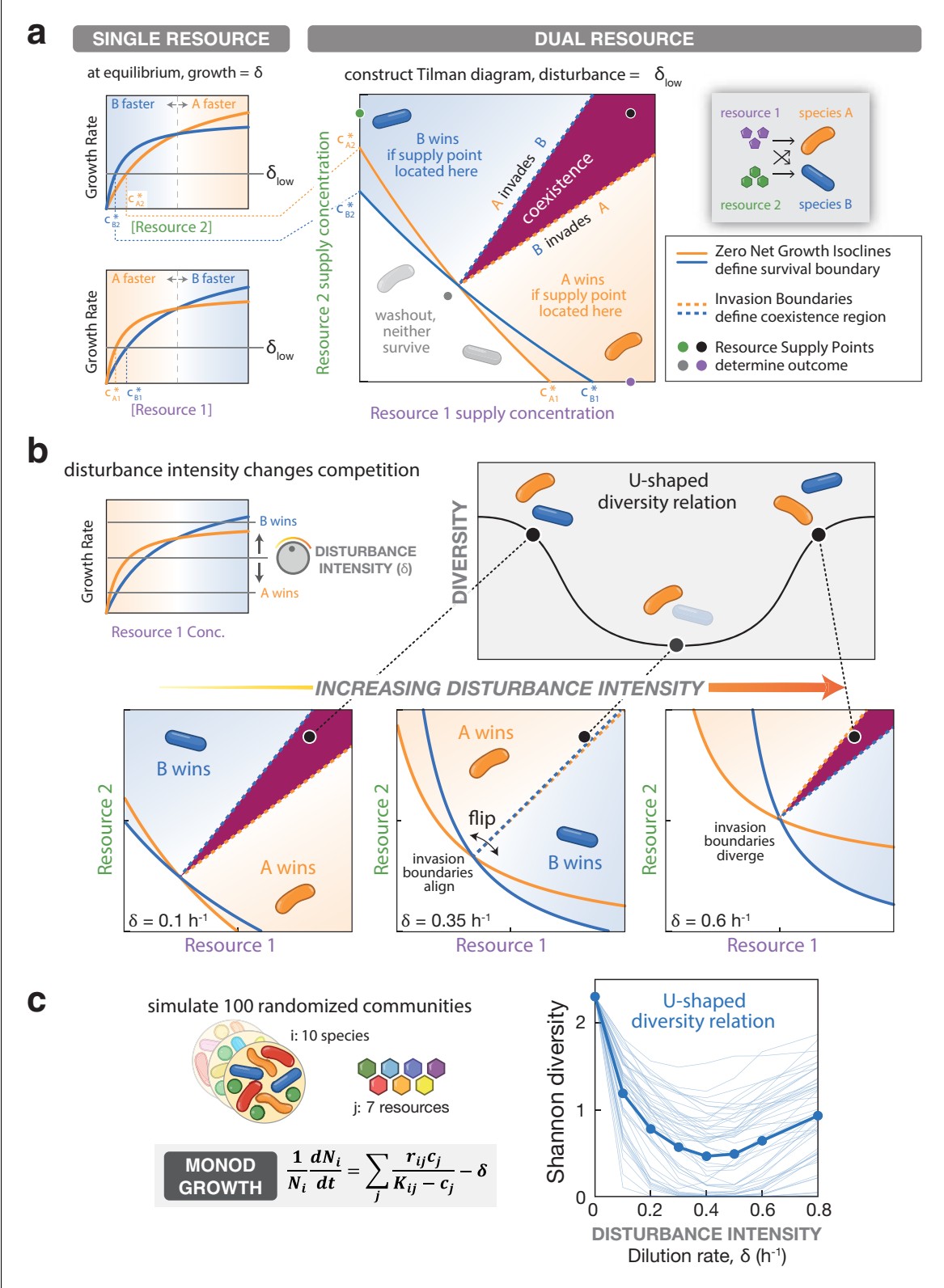

**Figure 2.** A U-shaped diversity disturbance relationship can emerge from a consumer resource model if species undergo niche-flip as disturbance increases. (a) The survival of a species in a consumer resource model depends on the supplied resource levels and the mortality rate (i.e. disturbance intensity). A Zero Net Growth Isocline (ZNGI) may be defined for each species at a given disturbance intensity, delineating the range of resource supply levels where growth can meet or exceed the mortality rate. The ZNGIs intersect the axis at c* values, the minimum concentration needed to support

*Figure 2 continued on next page*

*Figure 2 continued*

growth of a species on a single resource. The outcome of competition depends on the combination of resources supplied, indicated by a supply point. Resources are consumed until reaching equilibrium along the ZNGI of the species requiring the fewest resources to survive at the specified mortality rate, the winner of the competition. Different supply points can be found that support neither species (gray), a single species (green or purple) or both species (black). Invasion boundaries indicate regions where one species can increase in density in the presence of the other, outlining the region of coexistence (maroon). (b) In a consumer resource simulation with 2-species / 2-resources and Monod growth, the outcome of competition at the given supply point (black) depends on disturbance intensity. Both ZNGIs and invasion boundaries flip as disturbance intensity increases. At intermediate disturbance, invasion boundaries align and the coexistence region collapses, reducing diversity relative to low/high disturbance intensities at the black supply point. (c) Left: Monod consumer resource model for growth of species i with additive non-linear growth on each resource. Right: Shannon diversity of randomly generated 10-species communities, after six days of simulated growth on seven resources at varying dilution rates. For each model, mean diversity was computed for 100 randomly initialized communities, across each mean dilution rate, 50 of which are shown as individual traces.

The online version of this article includes the following figure supplement(s) for figure 2:

**Figure supplement 1.** Navigating between Monod growth curves and Tilman diagrams.

resource levels increase. For example, for two species competing for resource 1, species A may out-grow species B at low resource concentrations but not at high concentrations (*Figure 2a*). Competitive outcomes at low disturbance intensities will depend on the relative growth rates at low resource concentrations, while competitive outcomes at high disturbance intensities will depend on the relative growth rates at high concentrations (*Figure 2b*). The c* values and ZNGIs will change accordingly for each species. We investigated whether these disturbance-linked differences in competitive outcomes could explain the emergence of U-shaped DDRs. In a competition between two species, a U-shaped DDR can be generated if this coexistence region disappears at intermediate disturbance intensities (*Figure 2b*). We propose that this is possible if the ZNGIs and invasion boundaries flip as disturbance intensity increases, such that at some intermediate intensity the invasion boundaries align and the coexistence region disappears (*Figure 2b*). We term this behavior 'niche-flip'. Under niche-flip, the winner of competition at a given supply point changes as disturbance intensity varies concomitantly with resource availability.

To look for conditions under which niche flip emerges, we conducted simulations using Monod growth kinetics (*Monod, 1949*), a commonly used model for microbial growth with Type II functional response. Here, growth scales with the concentration of resource, but saturates to a maximal growth rate $r$ according to a half-saturation constant $K$ (*Figure 2c* and Appendix 1). Accordingly, a species with high maximum growth rate $r$ may be outcompeted at low resource levels by a species with a low saturation constant $K$, such that the outcome of competition varies depending on nutrient levels (and thus dilution rate $\delta$) as described above (*Figure 2—figure supplement 1*). Theoretical analysis of the Monod model with two species and two resources shows that ZNGIs and invasion boundaries undergo niche-flip as dilution rate increases (*Figure 2b*). We next asked whether these interactions between pairs of species produced the desired DDRs in a more complex system. We simulated sets of 10 species and seven resources with randomly chosen $r$ and $K$ values, with per-capita growth rates composed of a sum across nutrient-specific growth rates. Excitingly, we found that the Monod consumer resource model recapitulates the U-shaped diversity dependence on disturbance intensity that we observed in our chemostat experiments (*Figure 2c*). Finally, we emphasize that the illustrated tradeoff is not the only way to obtain niche-flip and U-shaped DDRs (explored further in Appendix 2). To summarize briefly, in the absence of tradeoffs on any one resource, at sufficiently high dilution rates, niche flip and a U-shaped DDR can arise from combinations of two or more resources such that the winner of competition is different at low vs. high equilibrium resource concentrations. In systems of larger numbers of species and resources, niche-flip exclusion events between pairs of species can co-occur, yielding U-shaped DDRs (Appendix 2).

## Other classes of microbial competition model do not produce U-shaped DDRs

We asked whether niche flip or U-shaped DDRs could emerge from other classes of microbial competition model. We simulated 10-species, 7-resource competitions with linear consumer resource models where the growth of a species increases linearly with resource concentration (Appendix 1). We also simulated 10-species competitions with the Lotka-Volterra model, where the maximal

growth rates ($r$) of different species are reduced depending on the abundance of other species via interaction coefficients ($\alpha$) (Appendix 1). Growth rates and interaction coefficients were selected to match the growth rates and diversity of the consumer resource models described above. U-shaped DDRs were not observed for either linear consumer resource or Lotka-Volterra models; instead, diversity generally did not vary with disturbance intensity, even under alternative parameter regimes (*Appendix 1—figures 1* and *2*). Thus, we concluded that the Monod consumer resource model best captured the U-shaped behavior of our microbial communities under constant disturbance among simple models of competition.

## Experimentally measured growth parameters are consistent with Monod growth

To confirm that Monod growth is a reasonable choice for modeling our experimental system, we measured how nutrient concentrations affected the growth of bacterial isolates from our community experiments. In 96-well plates, we first measured growth rates at varying concentrations of diluted Nutrient Broth, the same media type used in our eVOLVER experiments. Growth rates varied as a function of nutrient composition and indeed saturated at relevant nutrient concentrations, exhibiting tradeoffs between $r$ and $K$ such that no species had both a fast growth rate and low saturation coefficient (*Figure 3—figure supplement 4*). Second, as Nutrient Broth is undefined, we repeated the above experiment in carbon-limited M9 minimal media supplemented with varying concentrations of amino acids as a sole carbon source. Although we were unable to fit Monod parameters due to limited growth, we observed that the isolate with highest growth rate varied for different amino acids (*Figure 3—figure supplement 4*). These two experimental observations provide evidence for Monod growth kinetics under these conditions, and suggest that species could undergo niche-flip to generate U-shaped DDRs under constant disturbance.

## The consumer-resource model predicts fluctuation-dependent changes in DDRs

Both mean disturbance intensity and fluctuations (e.g. low-frequency discrete disturbance events) are hypothesized to play a role in the assembly of communities, but how these two disturbance components interact to reshape DDRs is unclear. Using our modeling framework, we sought to independently vary these two components, simulating a two-dimensional dilution profile. Specifically, we introduced fluctuations into the model by permitting $\delta$ to vary with time, compressing disturbance into discrete time windows (*Figure 3a*); this was done while keeping the time-averaged $\delta$ equal, thereby allowing us to vary mean disturbance intensity and frequency independently. The Monod consumer resource simulations predict significantly higher diversity in fluctuating conditions comprised of one or more dilution events per day, with the lowest-frequency (i.e. largest-fluctuation) regime predicted to maintain the most diversity (*Figure 3e* and *Figure 3—figure supplement 5*). This is consistent with intuition that fluctuations introduce temporal structure into environments, which may create new niches that promote diversity. Furthermore, the DDR is reshaped entirely – from U-shaped to largely uniform – indicating that community composition in the Monod model is conclusively fluctuation-dependent. Notably, neither Lotka-Volterra nor linear consumer resource models predict differences in the DDR between fluctuation frequencies (*Appendix 1—figures 1* and *2*). The overlap of DDRs of different frequencies indicates that in these models the relevant metric is the time-averaged overall intensity, rather than the frequency, of disturbance (*Chesson and Huntly, 1997*; *Fox, 2013*).

## Experimentally introducing disturbance fluctuations increases community diversity and reshapes DDRs

We returned to experiments to see whether the U-shaped DDR observed in constant dilution chemostats is reshaped by fluctuations, as predicted by the Monod model. To comprehensively test for fluctuation-dependency, we implemented dilution profiles with 1, 4, or 16 fluctuations per day (alongside the constant dilution conditions) at varying mean dilution rates in an eVOLVER experiment comprised of 64 simultaneous cultures ('DDR64 Experiment') (*Figure 3b*). We cultivated replicate samples of the soil-derived microbiome in eVOLVER for 6 days, taking samples every 24 hr to quantify composition (*Figure 3—figure supplement 1*, *Figure 3—source data 1*). The specific

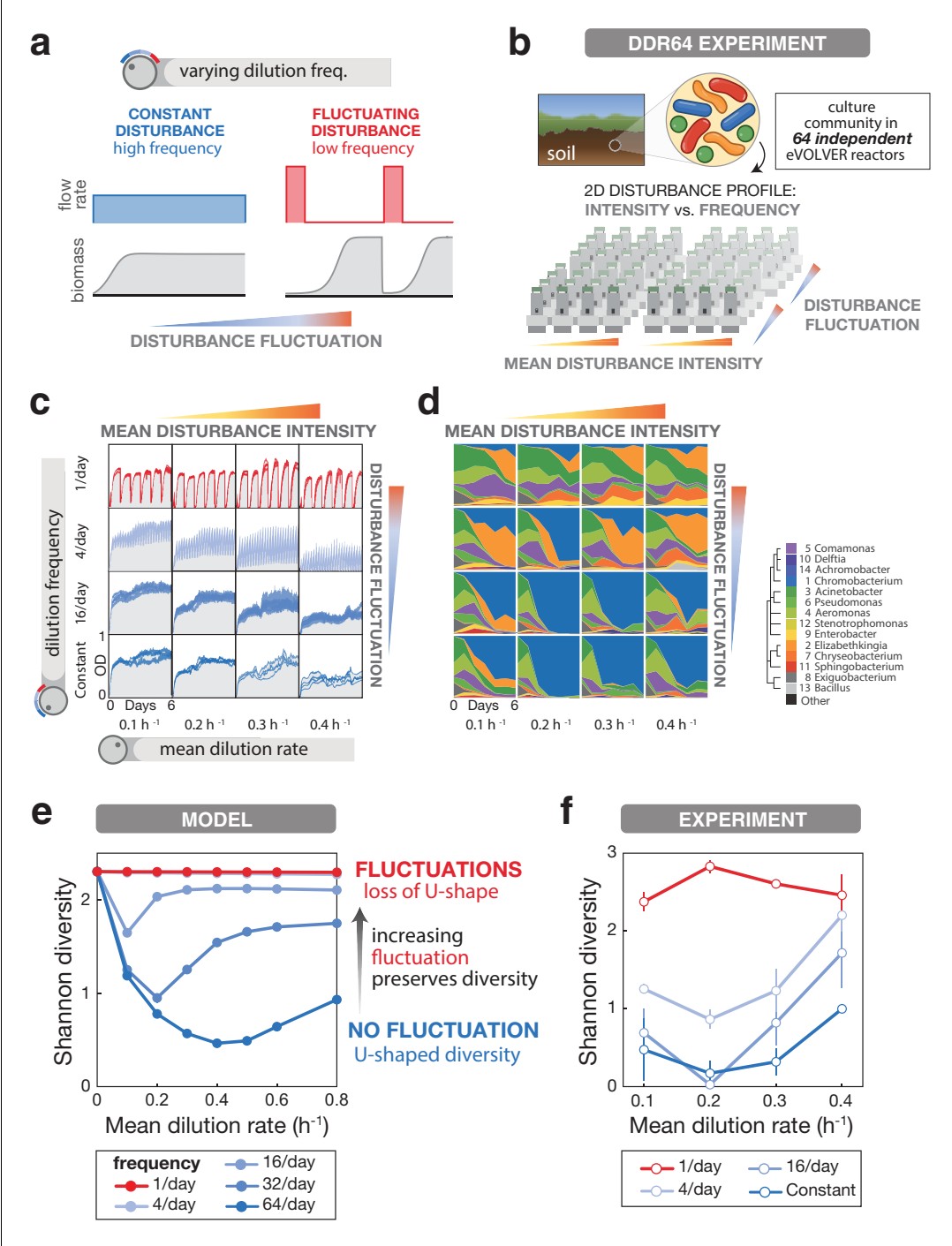

**Figure 3.** Introducing environmental fluctuations reshapes DDR and increases diversity levels in a microbial community. (**a**) Fluctuations in a disturbance over time cause fluctuations in biomass, and can be varied independently of the disturbance intensity. In continuous culture, fluctuations are achieved by aggregating dilution into discrete events while keeping mean dilution rate constant per day. (**b**) Schematic of the eVOLVER DDR64 experiment in which disturbance components (intensity and fluctuation) are varied independently. Samples of a soil-derived bacterial community were continuously cultured for 6 days across 64 eVOLVER bioreactors with varying mean dilution rate and dilution frequency. (**c**) Optical density traces for culture replicates in each condition show the dependence of biomass on disturbance. (**d**) Mean relative abundance of bacterial genera from replicate cultures, determined by 16S sequencing. Mean rank abundance is denoted by order, taxonomic similarity is denoted by color (see legend). (**e**) Mean Shannon diversity across 100 Monod consumer resource model simulations with varying mean dilution rate and dilution frequency show that the dependence of diversity on disturbance is fluctuation-dependent. (**f**) Mean Shannon diversity of Amplicon Sequence Variants from the DDR64

*Figure 3 continued on next page*

*Figure 3 continued*

experiment vs. mean dilution rate and dilution frequencies. As in the simulations, fluctuations increase diversity and eliminate the U-shape. Bars in e and f denote standard error of the mean.

The online version of this article includes the following source data and figure supplement(s) for figure 3:

**Source data 1.** Population size and diversity metrics for DDR64 and DDR washout.
**Source data 2.** Genus level composition for DDR64 and DDR washout.
**Figure supplement 1.** Timeline of DDR64 and DDR washout experiments.
**Figure supplement 2.** Inoculum composition determined by 16S sequencing.
**Figure supplement 3.** Optical density over time for DDR64 and DDR washout experiments.
**Figure supplement 4.** Measured growth of isolates in Nutrient Broth indicate variation in r and K parameters across different medias.
**Figure supplement 5.** Species richness from Monod Consumer Resource model exhibits U-shaped DDRs.
**Figure supplement 6.** Correlation between population size estimates vary according to condition and biofilm accumulation.
**Figure supplement 7.** Genus-level composition over time in the DDR64 and DDR Washout experiments.
**Figure supplement 8.** Principle coordinate analysis shows varying composition in DDR64 experiment.
**Figure supplement 9.** PERMANOVA analysis shows disturbance frequency significantly affects composition.
**Figure supplement 10.** Diversity changes over time in DDR64 and DDR washout experiments.
**Figure supplement 11.** Different diversity metrics from DDR64 and DDR washout experiments show U-shaped DDRs.
**Figure supplement 12.** Heatmap of biofilm levels at endpoint.
**Figure supplement 13.** Diversity does not correlate with potential confounding factors.

dilution profiles we programmed were reflected in the optical density traces of each culture over time, showing differences between conditions but close agreement between replicates (*Figure 3c* and *Figure 3—figure supplement 3*). Although optical density is a poorly defined population size metric for communities of different compositions, cultures with higher optical densities did tend to have larger population sizes measured by CFU/mL or DNA extraction (*Figure 3—figure supplement 6*). Based on 16S sequencing (*Gohl et al., 2016*; *Bolyen et al., 2019*; *Callahan et al., 2016*; *Janssen et al., 2018*), we observed that the genus-level composition of the community varied over time and between conditions (*Figure 3d* and *Figure 3—figure supplements 7* and *8*, *Figure 3—source data 2*). Culture compositions diverged from the initial composition, and while it is difficult to confirm that the endpoint represents true equilibrium (stable composition), the change in composition slows significantly by days 4–6, suggesting the endpoint state is approaching equilibrium. Principal Coordinate Analysis revealed that constant dilution conditions and 1/day fluctuations diverged from each other, indicating a clear frequency-dependent effect, with the spread potentially modulated by mean dilution rate (*Figure 3—figure supplement 8*). PERMANOVA statistical analysis of endpoint compositions confirmed that dilution frequency (but not mean dilution rate) had a significant effect on composition, but not mean dilution rate (*Figure 3—figure supplement 9*). Despite separation between conditions in PCoA of endpoint compositions (*Figure 3—figure supplement 9*), PERMANOVA analysis of dilution rate and frequency combinations did not yield significant values after correcting for false discovery rate. Notably, despite starting from a diverse community with hundreds of species, we found resulting compositions to be largely similar between replicates (*Figure 3—figure supplements 2* and *7*).

We calculated Shannon diversity for each timepoint (*Figure 3—figure supplement 10*, *Figure 3—source data 1*) and found that endpoint diversity trends across disturbance intensity and fluctuation frequency are qualitatively consistent with the Monod consumer resource model in three ways (*Figure 3e and f*, and *Figure 3—figure supplement 11*). (1) We observed U-shaped diversity curves in regimes of constant disturbance and small frequent disturbances, in both experiment and simulations. (2) Larger fluctuations preserved higher levels of diversity, and (3) larger fluctuations reshaped the diversity-disturbance curves towards a more uniform relationship. These general results were consistent across different diversity metrics: Shannon diversity of 16S ASVs (Amplicon Sequence Variants), richness of 16S ASVs, and richness of colony morphotypes (*Figure 3—figure supplement 11*).

To test the reproducibility of our results, we ran a subsequent and independent 48-vial experiment from frozen inoculum at dilution rates ranging from 0.3 $h^{-1}$ to 1.5 $h^{-1}$ (*Figure 3—figure supplements 1* and *3*). In overlapping parts of the parameter space, we were able to reproduce the results of the DDR64 experiment (*Figure 3—figure supplements 7*, *10* and *11*). We further used this replicate experiment to extend the parameter space and examine effects at extreme levels of

disturbance. However, despite imposing dilution rates that exceed the maximum growth rate of our cultures (*Figure 3—figure supplement 4*), we did not observe complete washout (*Figure 3—figure supplement 3*). In principle, cells could escape dilution by forming biofilm or due to incomplete mixing at extreme dilution rates, so we were cautious to draw strong conclusions from this part of the parameter space.

Although other measurables varied across the disturbance parameter space, they do not explain the differences in diversity as clearly as the Monod consumer resource model does (*Figure 3—figure supplements 6*, *12* and *13*). For example, we measured differences in biofilm accumulation across the parameter space using optical density measurements in eVOLVER, but these did not correlate with diversity. Similarly, changes in pH, oxygenation, phage, or antimicrobial levels could conceivably be linked to diversity changes and were not measured in this experiment. Notably, modifying the Monod consumer resource model to include asymmetric mortality rates did not qualitatively change the shape of DDRs in simulations (*Appendix 1—figure 4*), suggesting that our results could be robust to these types of effects. Overall, we found it striking that the model captures the features of our results so well while being relatively simple and non-parameterized.

## The consumer-resource model can produce all major classes of DDRs and is robust to alternative formulations

We set out to develop a conceptual framework that not only captures our results, but also provides mechanistic intuition about how the characteristics of disturbance, such as frequency, can produce and reshape DDRs more generally. In this case, we demonstrated how a U-shaped DDR emerging

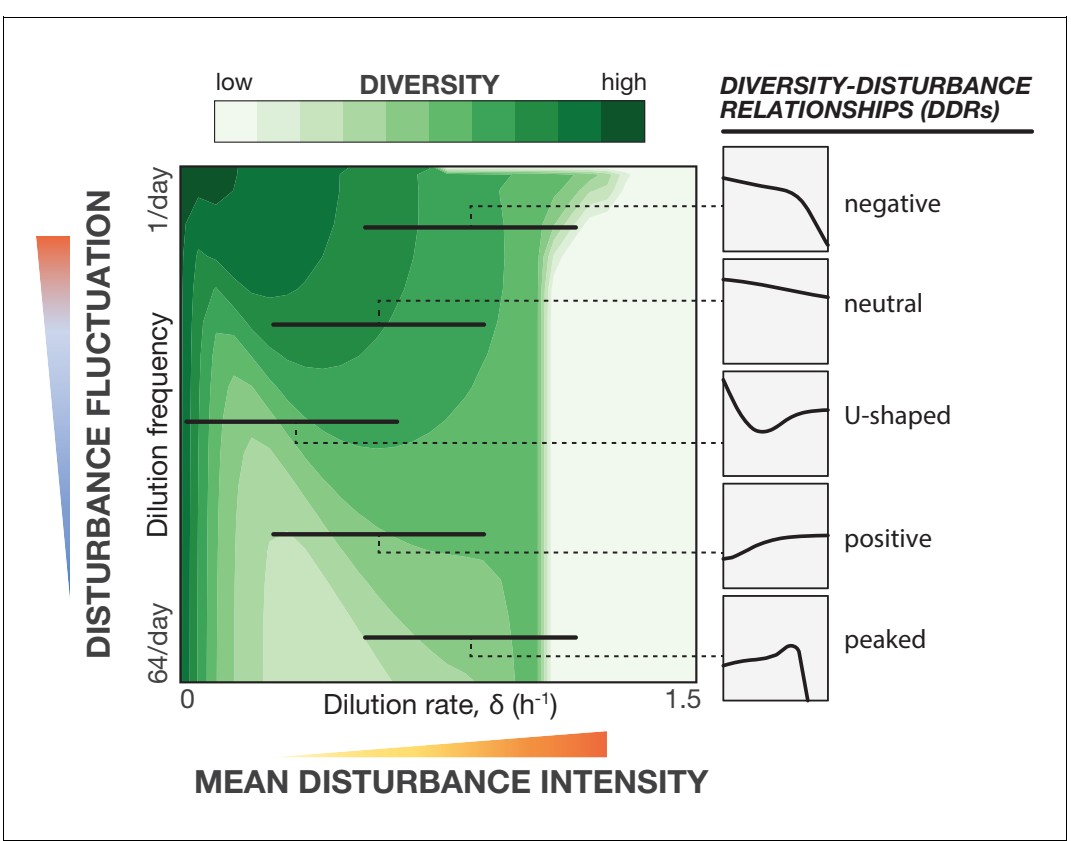

**Figure 4.** Distinct classes of diversity-disturbance relationships emerge when traversing different regions of disturbance (intensity vs. fluctuation) space. A simple Monod consumer resource model maps a phase diagram for community diversity by varying distinct characteristics of environmental disturbance (intensity and fluctuation). Diversity varies non-linearly with disturbance intensity under different levels of fluctuation for Type II consumer resource models. Contour plot depicts Shannon diversity results for a Monod consumer resource model simulation as an illustrative example. When varying disturbance intensity at a fixed fluctuation level along the indicated arrows, different DDR classes emerge, as shown in smoothed plots on the right.

under constantly-applied disturbance was reshaped to a more neutral relationship by the addition of low-frequency fluctuations. This framework further identified a niche-flip mechanism as a dominant factor in community assembly under constantly applied disturbance. We next wondered whether this simple model could produce other classes of DDRs observed in nature (*Mackey and Currie, 2001*), as a path toward identifying the organizational mechanisms in other ecosystems. To explore a wider range of behaviors, we extended the range of simulations to include extreme disturbance intensities that eventually lead to community extinction, and prevented artificial extinction of all species simultaneously by introducing noise into the normalization of growth rates (creating small differences in relative fitness between species). As depicted in a contour plot (*Figure 4*), we again found that fluctuations reshaped the DDRs to produce complex diversity landscapes. These simulations revealed that a diverse repertoire of DDRs can emerge from the combination of these two disturbance characteristics (intensity and frequency). By exploring subsets of this parameter space, we observed every major class of DDR: positive, negative, neutral, peaked, and U-shaped (*Figure 4*). Different behaviors can emerge from similar communities maintained under different portions of the disturbance parameter range, or when similar disturbances are applied to communities with different growth parameters. By simply examining these two disturbance characteristics in a systematic way, we developed a unifying framework that can reconcile disparate DDR observations without invoking more complex phenomena.

To check how well these behaviors were preserved under variations of the Monod model, we performed simulations in which we (1) modestly varied parameter values, (2) significantly changed parameter distribution assumptions, or (3) changed the functional form of the model altogether. Notably, as detailed below, the U-shaped DDR and frequency-dependence were maintained for nearly all model variations, indicating that disturbance intensity and fluctuations remained important predictive factors of community diversity.

First, we made quantitative changes to parameters that did not qualitatively change the relationship between species. In one example, we explored larger relative fitness differences between species by removing growth rate normalization between species, but the U shape and fluctuation-dependent DDR changes were preserved (*Appendix 1—figure 3e*). In another, we chose parameters in ranges that diverged from experimental conditions and found the model results to be robust to increased simulation length (*Appendix 1—figure 3d*) and 10-fold increased resource supply (*Appendix 1—figure 3b*), but not 10-fold decreased resource supply (*Appendix 1—figure 3c*). At increased number of species and resources, a U-shaped DDR is still observed, although its relative impact on total diversity is reduced (*Appendix 1—figure 3i*).

Second, we modified parameters in ways that changed the underlying assumptions about community structure. We found no qualitative changes to model results when including 'specialist' species or eliminating $r/K$ tradeoffs with sorted $r$ and $K$ values (*Appendix 1—figure 3f and g*). However, when we held $K$ constant for all species on each resource, both the U shape and frequency dependence were eliminated (*Appendix 1—figure 3h*). For a mathematical discussion of the importance of these parameter choices, see Appendix 2.

Finally, we sought to determine whether changing the functional form of the summed Monod model would affect the U shape. First, we employed a non-additive formulation of Monod growth on mixed substrates (*Sears and Chesson, 2007*); similar U-shaped DDRs and frequency dependence were observed (*Appendix 1—figure 4*). Next, we incorporated the possibilities of migration or species-specific death rates in different models. These modifications respectively increased or decreased the maximum diversity of the system, but the U-shaped DDR and frequency-dependence were maintained (*Appendix 1—figure 4*). Again, we found it striking that over a wide range of model variations, disturbance intensity and fluctuations remained important factors shaping the diversity of ecosystems in a predictable manner.

## Discussion

In this work, advances in automated continuous culture technology and next-generation sequencing enabled laboratory microbial ecology studies to systematically dissect the role of environmental disturbance (intensity and fluctuation) with fine resolution at scale. We found replicable patterns in composition and diversity of a soil-derived microbial community across different disturbance regimes. Notably, we observed an unexpected U-shaped DDR under constant disturbance, and

found that adding fluctuations (in the form of low-frequency discrete disturbance events) increased community diversity and reshaped the DDR. All of these results are well captured by the Monod consumer resource model, which subsequently led us to describe and propose a novel niche-flip mechanism for structuring these ecosystems. Taken together, these combined experimental and modeling results (1) provide new insight into how microbial community assembly depends on environmental conditions and (2) demonstrate a role for environmental fluctuations in promoting diversity. Examining the behavior of the Monod model over a broad range of disturbance intensities and frequencies, we see that diverse classes of DDRs (including increasing, decreasing, and peaked) could emerge when only subsets of the parameter space are sampled (*Figure 4*). Understanding how disturbance intensity and frequency interact is essential for reconciling disparate observations of DDRs under a single quantitative unifying framework (*Mackey and Currie, 2001*; *Hughes et al., 2007*; *Hall et al., 2012*; *Miller et al., 2011*).

While we focused on two characteristics, a number of different temporal features of a disturbance could be considered (*Mackey and Currie, 2001*; *Hall et al., 2012*; *Miller et al., 2011*). In our present work, by focusing on mean disturbance intensity rather than maximum disturbance intensity, we have sampled conditions with a negative correlation between frequency and maximum intensity (e.g. comparing frequent small dilution events to infrequent large dilution events). Furthermore, our definition of constant disturbance may in fact represent the undisturbed state of some ecosystems. For natural ecosystems with constant flow, such as lentic aquatic systems, it may be more accurate to consider a drought disturbance with a sudden reduction in flow rate (with distinct biological outcomes). Performing an experiment in which the maximum flow rate, minimum flow rate, duration at maximum and duration at minimum were all independently varied could help to understand a wider range of disturbance profiles.

With further study and evaluation of underlying assumptions, the findings of this work may be extended to other systems. Although our experimental results were reproducible (*Figure 3—figure supplements 7*, *10* and *11*), it remains to be seen whether other species and disturbance types (including asymmetric disturbances like toxins or heat shock) behave similarly in experiments. Our simulation findings indicate that the niche-flip mechanism is quite generalizable across varying assumptions and formulations, and thus would predict similar roles for other disturbance types and conditions. Furthermore, the growth saturation feature of the Monod model that enables niche-flip is also characteristic of Type II and III functional responses used in consumer resource models across different scales of ecology. It remains to be seen whether niche-flip mechanisms could arise from non-resource-based models. It is plausible that similar growth tradeoffs arising in response to other disturbance-correlated features could lead to loss of coexistence at intermediate disturbance intensities. Therefore, niche flip could be a more general principle extending beyond relative growth nonlinearities explored in this work to systems driven by dynamic abiotic stresses and/or storage effects (*Sears and Chesson, 2007*; *Angert et al., 2009*; *Usinowicz et al., 2017*; *Hallinen et al., 2020*). We also note however, that these types of disturbances do not share the direct link between environmental change and biological outcome that is characteristic of dilution disturbance, so the impact may be less clear.

Broadly, our results highlight that the structure of ecosystems and their response to perturbation is contextual. We demonstrated that increasing the disturbance intensity can increase, decrease, or have no effect on the diversity of a system. Critically however, we found that under a unifying framework that considers both disturbance intensity and frequency, these relationships become predictable rather than idiosyncratic (*Hall et al., 2012*; *Miller et al., 2011*). Intriguingly, these complex ecosystem assembly rules can emerge from temporal structure alone, without invoking other organizing principles, such as spatial structure (*Gude et al., 2020*) or network structure (e.g. cross-feeding and antagonism) (*Mee et al., 2014*; *Kelsic et al., 2015*). While we considered both constant and oscillating temporal patterns of disturbance, additional complexity may arise from temporal patterns that are random or those which change which types of resources available over time. Since complexity can arise from temporal, spatial, and network effects, the relative importance of any one organizing factor is likely ecosystem dependent. If predictable response to perturbation depends on context, then designing predictable interventions to ecosystems (in medicine, agriculture, and conservation) will require the ability to measure and understand the environmental context. With the staggering amount of compositional data being generated with high-throughput sequencing of

microbiomes (*Thompson et al., 2017*), inference of environmental context and design of robust ecological interventions may not be far off.

## Materials and methods

### Preparing inoculum

Two g of dirt from the Communications Lawn of Boston University (collected on 09/15/2018) was vortexed in 10 mL PBS + 200 µg/mL cycloheximide, then incubated in the dark at room temperature for 48 hr. For pre-enrichment, 16 eVOLVER vials were prepared with 25 mL of 0.1X Nutrient Broth (NB) media (0.3 g/L yeast extract + 0.5 g/L peptone (Fisherbrand)) with 200 µg/mL cycloheximide, inoculated with 350 uL of PBS immersion, and grown for 18 hr in eVOLVER at 25°C. All 16 pre-enrichment cultures were mixed together to form the experiment inoculum. Aliquots in 15% glycerol were stored frozen at −80°C.

Running eVOLVER Experiments eVOLVER lines were sterilized using 10% bleach and ethanol (*Heins et al., 2019*; *Wong et al., 2018*), then autoclaved vials were loaded with 23 mL of 0.1X NB. Each vial was inoculated with 1 mL of inoculum, and grown for 5 h at 25°C with stirring prior to the first dilution disturbance. eVOLVER was operated in chemostat mode with 0.5 mL bolus size, with dilutions either evenly distributed over time (constant disturbance) or concentrated in fluctuation periods lasting 15 minutes. For these cultures, the flow rate during a fluctuation $\delta_f$ depended on the number of fluctuations per day $f$ and mean dilution rate $\delta$ according to the following equation: $\delta_f = (24 * \delta)/(0.25 * f)$. At the end of each experiment, vials were flushed with media, and 10 optical density measurements were taken in eVOLVER to measure the biofilm levels.

Bottles and lines were routinely checked for contamination. This occurred to only a single vial of the experiment, which was excluded from statistical analysis. For the follow-up washout experiment, the glycerol stock inoculum was thawed at room temperature, 1 mL was inoculated into each vial, then the cultures were allowed to grow for 5.7 hr prior to initiating disturbances. For the washout experiment, a software bug caused a few incorrectly executed dilution events; these vials were excluded from statistical analysis. Code required to execute these experiments will be available on Github (https://github.com/khalillab/DDR-eVOLVER-model).

### Sampling cultures

At each timepoint, a 2 mL culture aliquot was removed from each vial with an extended length pipette tip.

For plating, 20 µL of the sample was used for a 10-fold serial dilution series, and 100 uL of diluted cultures at three concentrations were plated on 18 mL Nutrient Broth Agar plates (3 g/L yeast extract, 5 g/L peptone, 15 g/L agar [Fisherbrand]), which were grown at room temperature for 48–60 hr, then imaged on an on an Epsom Perfection 550 scanner. Image analysis was performed with the aid of Cellprofiler 3.1.8 (*Lamprecht et al., 2007*) and Cellprofiler Analyst 2.2.1 Classifier (*Bray et al., 2015*) tools.

For DNA extraction, the remainder of the sample was pelleted and frozen at −20°C. 60–72 hr after freezing, pellets were lysed at 37°C for 1 hr in 200 uL of lysozyme buffer (25 mM Tris HCl pH 8.0, 2.5 mM EDTA, 1% Triton X-100 with 20 mg/mL lysozyme (Fisher), prepared fresh daily). Lysates were processed using DNEasy Blood and Tissue Kit according to manufacturer specifications, eluted into 10 mM Tris buffer, and normalized to 5 ng/µL DNA based on measurements in a Qubit fluorometer.

### Library preparation and sequencing

Briefly, we performed amplicon sequencing of the 16S v4 region based on established protocols (*Gohl et al., 2016*). Primers prCM543 (TCGTCGGCAGCGTCAGATGTGTATAAGAGAC-AGGTG YCAGCMGCCGCGGTAA) and prCM544 (GTCTCGTGGGCTCGGAGATGTG-TATAAGAGACAGG-GACTACNVGGGTWTCTAAT), adapted from EMP515F (*Apprill et al., 2015*) and EMP806R (*Parada et al., 2016*) were used to isolate a 290 bp 16S v4 region, using Kapa Hifi ReadyMix polymerase and the following cycling conditions: (i) denaturation: 95°C for 5 min; (ii) amplification (25 cycles): 98°C for 20 s, 55°C for 15 s, 72°C for 1 m; (iii) elongation: 72°C for 5 min. For the negative control and biofilm samples, the number of cycles was increased to 35 to amplify from low biomass.

Illumina NexteraXT primers (or equivalents) were used to form a final library 427 bp in length, with the following conditions: (i) denaturation: 95℃ for 5 min; (ii) amplification (eight cycles): 98℃ for 20 s, 55℃ for 15 s, 72℃ for 1 m; (iii) elongation: 72℃ for 10 min. DNA was purified with AMPure XP beads or SequalPrep plates, then samples were multiplexed in groups of 192 alongside control samples at a higher fraction, and spiked with PhiX or whole genome DNA libraries to a final concentration of 50% to increase sequence diversity. Library pools were sequenced at the Harvard Biopolymers Facility across five 250 bp paired end MiSeq v2 runs.

## Sequencing analysis

Samples were demultiplexed using the Illumina BaseSpace demultiplexer analysis tool. All subsequent bioinformatic analysis was performed in QIIME2 v2020.2 (*Bolyen et al., 2019*). Demultiplexed samples were dereplicated using DADA2 sample inference to tabulate Amplicon Sequencing Variants (ASVs) (*Callahan et al., 2016*). Next, for qualitative description of composition, taxonomy (to the genus level) was assigned to each feature by alignment to the SILVA 132 database (*Quast et al., 2013*) using the taxa-barplot plugin. For quantitative analysis, samples with technical issues (e.g. contamination, low biomass, poor sequence quality, etc.) were removed and the remaining 698 samples were rarefied to 6840 reads. The fragment-insertion plugin was used to generate a rooted phylogenetic tree using the SEPP algorithm (*Janssen et al., 2018*). The diversity plugin was used to calculate Shannon diversity, ASV richness, and weighted UniFrac distance (*Lozupone et al., 2011*), which was used to perform Principle Coordinate Analysis (PCoA) and PERMANOVA analysis.

## Isolating strains

Colonies from imaging plates for the inoculum and endpoint timepoints of the DDR64 and follow-up washout experiments were re-struck on NB agar plates, grown overnight in 0.1X NB media, and stored frozen at −80℃ in 15% glycerol. Primers prCM215 (CCATTGTAGCACGTG-TGTAGCC) and prCM216 (ACTCCTACGGGAGGCAGC) were used to amplify the v3-v7 16S region for Sanger sequencing and identification.

## Growth characterization of isolates

Several strains with different taxonomic background were selected from the collection of isolates for further study. For measurement of $r/K$ values, nine strains were struck onto NB plates, and single colonies were grown to stationary phase overnight at 30℃, then diluted to an OD600 of 0.001 in triplicate 200 uL cultures in dilute NB media (ranging from 0.1X to <0.0016X, plus a water-only negative control) in 96-well microplates grown in a Tecan spectrophotometer for 36 hr at 30℃, shaking, with lid on. Growth rates were fit to log transformed OD600 data, Lineweaver-Burke plots were constructed across media concentration, and Monod curves were fit to each species. Three strains that could grow in M9 defined minimal media were selected for characterization on different limiting resources. Strains were struck onto NB plates, and single colonies were grown to stationary phase overnight at 30℃ in carbon-limited M9 supplemented with aspartate, glutamate, or proline, then diluted and grown in microplates as above, with amino acid concentrations ranging from 10 mM to <0.16 mM, plus a no-carbon control. Due to limited growth on these media, optical densities at sub-saturating resource concentrations were below the limit of detection, so only maximum growth rate at 10 mM amino acid was reported.

## Data and materials availability

All sequencing data is deposited in the Sequence Read Archive (SRA) accessible with a BioProject accession code PRJNA719465. Agar plate images are being deposited on Figshare accessible at https://doi.org/10.6084/m9.figshare.15117558. Computer code used to run eVOLVER experiments and for theoretical modeling is available at https://github.com/khalillab/DDR-eVOLVER-model (copy archived at swh:1:rev:7019f031598169723a5828f3909eba1e199794d2; *Khalil, 2021*). All other datasets required to produce the results in the current study are included as supplemental data.

## Acknowledgements

We thank J Goldford, D Segrè, and members of the Gore and Khalil groups for helpful discussions. We thank S Boswell for sequencing advice, M Springer and J Galagan for reagents and equipment used in library preparation, and the Harvard Biopolymers Facility for their services. Funding: This work was supported by DARPA BRICS grants HR001115C0091 (JG and ASK) and HR001117S0029 (ASK), Simons Foundation grant 542385 (JG), and NIH grants R01GM102311 (JG) and R01EB027793 (ASK). ASK also acknowledges funding from the NIH Director's New Innovator Award (DP2AI131083) and NSF CAREER Award (MCB-1350949).

## Additional information

### Competing interests

Ahmad S Khalil: is co-founder of Fynch Biosciences, a manufacturer of eVOLVER hardware. The other authors declare that no competing interests exist.

### Funding

| Funder | Grant reference number | Author |
| --- | --- | --- |
| Defense Advanced Research Projects Agency | HR001115C0091 | Ahmad S Khalil |
| Defense Advanced Research Projects Agency | HR001117S0029 | Ahmad S Khalil |
| Simons Foundation | 542385 | Jeff Gore |
| National Institute of General Medical Sciences | R01GM102311 | Jeff Gore |
| National Institute of Biomedical Imaging and Bioengineering | R01EB027793 | Ahmad S Khalil |
| National Institutes of Health | DP2AI131083 | Ahmad S Khalil |
| National Science Foundation | MCB-1350949 | Ahmad S Khalil |

The funders had no role in study design, data collection and interpretation, or the decision to submit the work for publication.

### Author contributions

Christopher P Mancuso, Conceptualization, Software, Formal analysis, Investigation, Visualization, Methodology, Writing - original draft, Writing - review and editing; Hyunseok Lee, Clare I Abreu, Software, Formal analysis, Investigation, Methodology, Writing - original draft, Writing - review and editing; Jeff Gore, Conceptualization, Supervision, Funding acquisition, Writing - original draft, Project administration, Writing - review and editing; Ahmad S Khalil, Conceptualization, Supervision, Funding acquisition, Investigation, Writing - original draft, Project administration, Writing - review and editing

### Author ORCIDs

Christopher P Mancuso (iD) https://orcid.org/0000-0002-3974-3480
Hyunseok Lee (iD) http://orcid.org/0000-0003-1554-6228
Jeff Gore (iD) http://orcid.org/0000-0003-4583-8555
Ahmad S Khalil (iD) https://orcid.org/0000-0002-8214-0546

### Decision letter and Author response

Decision letter https://doi.org/10.7554/eLife.67175.sa1
Author response https://doi.org/10.7554/eLife.67175.sa2

## Additional files

### Supplementary files
• Transparent reporting form

### Data availability
All sequencing data is deposited in the Sequence Read Archive (SRA) accessible with a BioProject accession code PRJNA719465. Agar plate images are deposited on Figshare accessible at https://doi.org/10.6084/m9.figshare.15117558. Computer code used to run eVOLVER experiments and for theoretical modeling is available at https://github.com/khalillab/DDR-eVOLVER-model (copy archived at https://archive.softwareheritage.org/swh:1:rev:7019f031598169723a5828f3909eb-a1e199794d2). All other datasets required to produce the results in the current study are included as supplemental data. Source data files have been provided.

The following dataset was generated:

| Author(s) | Year | Dataset title | Dataset URL | Database and Identifier |
|---|---|---|---|---|
| Mancuso C | 2021 | DDR64 Petri Dish Photos | https://doi.org/10.6084/m9.figshare.15117558.v1 | figshare, 10.6084/m9.figshare.15117558.v1 |

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

## Appendix 1

### Comparison of simulation results under different microbial competition models

Here we describe the methods, equations, and parameter choices used in our simulations.

### Lotka-volterra simulations

We simulated 10-species competitions with the Lotka-Volterra model:

$$\frac{1}{N_i}\frac{dN_i}{dt} = r_i\left(1 - \sum_{k=1}^{10}\alpha_{ik}N_j\right) \tag{1}$$

$N_i$ represents the abundance of species $i$ modified by its carrying capacity (we rescale the carrying capacities of all species to one), $r_i$ represents its maximal growth rate, and $\alpha_{ik}$ represents inhibition of species $i$ by species $k$. The model is parameterized such that $\alpha_{ii} = 1$.

Simulations mimicked experimental conditions as much as possible. Beginning with equal abundances of all 10 species, we integrated equations for a six-day period using the function ode45 in MATLAB. Species abundances were diluted during equally spaced 15 min intervals by integrating a version of *Equation 1* modified to include dilution:

$$\frac{1}{N_i}\frac{dN_i}{dt} = r_i\left(1 - \sum_{k=1}^{10}\alpha_{ik}N_k\right) - \delta \tag{2}$$

$\delta$, the death or dilution rate, was calculated by distributing the mean dilution rate $D$ (per hour) over equally spaced intervals of 15 minutes (per 1/4 hour) at a frequency $f$ (per 24 hours):

$$\delta = \frac{1}{\left(\frac{1}{4}\right)hr} * \frac{D\left(\frac{1}{hr}\right)}{f\left(\frac{1}{24hr}\right)} = 96 * \frac{D}{f}\left(\frac{1}{hr}\right) \tag{3}$$

Frequency $f$ of dilution ranged from 1 to 64 per day, and mean dilution rate $D$ ranged from 0.1 to 0.8 per hour. Maximal growth rates $r_i$ were randomly sampled from a normal distribution with mean 1 and standard deviation 0.1. This range was selected to roughly match measured growth rates of isolates on 0.1X Nutrient Broth (*Figure 3—figure supplement 4*). Competition coefficients $\alpha_{ik}$ were randomly sampled from a lognormal distribution with parameters $\mu = -0.7$ and $\sigma = 0.2$, which are the mean and standard deviation of the associated normal distribution. The mean of the lognormal distribution is $\exp\left(\mu + \frac{\sigma^2}{2}\right) = 0.51$ and the standard deviation is $\exp(2\mu) + \sigma^2(\exp(\sigma^2) - 1) = 0.1$. The competition coefficients were selected to match the diversity of the resource-explicit simulation results at the zero-dilution condition (see below). Other ranges of $r_i$ and $\alpha_{ik}$ did not alter the DDR shape (*Appendix 1—figure 1*).

We simulated 100 competitions with randomly drawn parameters, across all dilution-frequency combinations. All 10 species began at equal abundances, and we used their final relative abundances $p_i$ to calculate the Shannon diversity index $\rho$ of each outcome:

$$\rho = \sum_{i=1}^{10} -p_i \ln p_i \tag{4}$$

In order to be counted, the abundance of a species $p_i$ had to exceed a threshold of 0.0001. Finally, we took the average of the 100 values of $\rho$ for each dilution-frequency combination to obtain average diversity.

### Linear consumer resource simulations

We simulated 10-species, seven-resource competitions with a linear consumer resource model:

$$\frac{1}{N_i}\frac{dN_i}{dt} = \sum_{j=1}^{7} r_{ij}c_j \tag{5}$$

$N_i$ represents concentration of species $i$, $r_{ij}$ its growth rate per unit resource on resource $j$, and $c_j$ the concentration of resource $j$. Units of species and resources per volume are the same, because we assume that one unit of resource is fully converted into one unit of biomass; this assumption is equivalent to assuming the biomass yield is equal to one for all species, and therefore we do not include a yield parameter.

Similar to the Lotka-Volterra simulations, we integrated the dilution-modified equations over the same time period and range of frequencies and dilutions:

$$\frac{1}{N_i}\frac{dN_i}{dt} = \sum_{j=1}^{7} r_{ij}c_j - \delta \tag{6}$$

$\delta$, as defined by *Equation 3*, is not only the dilution rate of cells, but also the influx of fresh resources at source concentration $c_{jo}$:

$$\frac{dc_j}{dt} = \delta(c_{jo} - c_j) - \sum_{i=1}^{10} N_i r_{ij}c_j \tag{7}$$

Simulations began with equal abundances of all species and resources ($c_{jo} = 1$), except for one resource, which had a slightly different supply concentration ($c_{jo} = 1.2$), in order to move the resource supply point away from a unique central position. Growth rates per unit resource $r_{ij}$ were randomly sampled from a normal distribution with mean 1 and standard deviation 0.1, and then normalized by dividing by the sum of growth rates for each species across all resources, $\sum_{j}^{7} r_{ij}$. Other parameter ranges did not alter the DDR shape (*Appendix 1—figure 2*).

## Monod consumer resource simulations

We simulated 10-species, 7-resource competitions with a Monod resource model:

$$\frac{1}{N_i}\frac{dN_i}{dt} = \sum_{j=1}^{7} \frac{r_{ij}c_j}{K_{ij} + c_j} \tag{8}$$

The Monod constant, $K_{ij}$, is the concentration of resource $j$ at which species $i$ reaches its half-maximal growth rate on that resource. Similar to the linear resource concentration model (*Equation 6 and 7*), the model can be modified to include dilution:

$$\frac{1}{N_i}\frac{dN_i}{dt} = \sum_{j=1}^{7} \frac{r_{ij}c_j}{K_{ij} + c_j} - \delta \tag{9}$$

$$\frac{dc_j}{dt} = \delta(c_{jo} - c_j) - \sum_{i=1}^{10} \frac{N_i r_{ij}c_j}{K_{ij} + c_j} \tag{10}$$

We performed simulations using the same procedure described above. The Monod constants $K_{ij}$ were randomly sampled from a uniform distribution: [0.001, 0.01]. The width of the selected range is consistent with the range of Monod constants measured on Nutrient Broth (*Figure 3—figure supplement 4*), and similar DDR shapes are generated with alternative parameter choices (*Appendix 1—figure 3*). Maximal growth rates $r_{ij}$ were sampled and normalized as described above. To check whether the U-shaped DDR was preserved under variations of the Monod model, we performed other simulations.

First, we introduced larger relative fitness differences between species would affect our results or yield different DDRs. We added Gaussian noise to maximal growth rates $r_{ij}$ with standard deviation of 0.02, which created small differences in relative fitness between species. We observed similar results, with a gradual decrease in the diversity as dilution rates exceed individual species maximal

growth rates (*Figure 4*). For larger relative differences, we removed normalization between species, and instead normalized maximal growth rates per resource by dividing by the number of resources, rather than the sum of growth rates across all resources (*Appendix 1—figure 3e*).

Diverging from the conditions of the experiment, we searched for alternative parameter ranges that could qualitatively alter the DDR. We found the model to be robust to increased simulation length (*Appendix 1—figure 3d*) and increased resource supply (*Appendix 1—figure 3b*). However, when reducing each of the resource supply concentrations, $c_{jo}$, by a divisive factor of 10, the U-shape was almost completely eliminated except for a small effect at the highest dilution frequency (*Appendix 1—figure 3c*). This modification is equivalent to increasing Monod constants, $K$, by a multiplicative factor of 10, placing them closer to the range of resource supply concentration, which does not reflect our measurements (*Figure 3—figure supplement 4*) and is less biologically relevant. This extreme parameter range essentially changes the model choice, moving the growth dynamics closer to those of the linear consumer resource model.

Next, we made modifications to parameter choices under the summed Monod model that change the underlying assumptions to see if any would eliminate the U shape. First, we included the possibility of "specialist" species by widening the maximal growth rate sampling range. We randomly sampled $r$ from a uniform distribution (0.1,1), before normalizing by dividing by the sum of growth rates for each species across all resources, $\sum_{j}^{7} r_{ij}$, as before. The U shape was preserved (*Appendix 1—figure 3f*). Next, we eliminated $r/K$ tradeoffs on all resources by sampling $r$ and $K$ as before, and then sorting them such that the species with the highest value of $r$ also had the lowest value of $K$, and the next-highest and next-lowest, and so on, for each resource. The U shape was preserved (*Appendix 1—figure 3g*). Finally, we sampled $K$ only once per resource, keeping it constant for all species on each resource. This modification eliminated the U shape (*Appendix 1—figure 3h* and *Appendix 2—figure 1*). For a mathematical discussion of the importance of these parameter choices, see Appendix 2.

Finally, we sought to determine whether changing the functional form of the summed Monod model would affect the U shape. First, we employed a non-additive formulation of Monod growth on mixed substrates (*Sears and Chesson, 2007*). In this formulation, the per-capita growth rate on multiple resources is a saturating function of Monod growth on individual resources, rather than a simple sum (as in *Equation 10* above):

$$\frac{1}{N_i}\frac{dN_i}{dt} = \frac{\lambda_c \sum_{j=1}^{7} \frac{\phi_{ij}}{\lambda_c - \phi_{ij}}}{1 + \sum_{j=1}^{7} \frac{\phi_{ij}}{\lambda_c - \phi_{ij}}} \tag{11}$$

Here, $\phi_{ij} = \frac{r_{ij}c_j}{K_{ij}+c_j}$, which is the Monod growth of species $i$ on resource $j$. The other parameter, $\lambda_c$, is the horizontal intercept in a plot of a species' catabolic enzyme expression as a function of its growth rate. For simplicity, we assumed $\lambda_c$ to be equal for all species. Because the species consume resources non-additively in this model, the resources are also depleted non-additively:

$$\frac{dc_j}{dt} = -\sum_{i=1}^{10} \frac{N_i \phi_{ij}(\lambda_c - \phi_{ij})}{\lambda_c - \frac{1}{N_i}\frac{dN_i}{dt}} \tag{12}$$

We can write the dilution-modified model as follows (substituting $\lambda_c \gamma_i$ for per-capita growth of species $i$ in the absence of dilution):

$$\frac{1}{N_i}\frac{dN_i}{dt} = \lambda_c \gamma_i - \delta \tag{13}$$

$$\frac{dc_j}{dt} = \delta(c_{jo} - c_j) - \sum_{i=1}^{10} \frac{N_i \phi_{ij}(\lambda_c - \phi_{ij})}{\lambda_c - \lambda_c \gamma_i} \tag{14}$$

Simulating this model with values of $\lambda_c$ ranging between 0.3 and 30 yielded DDRs similar to those of the summed Monod model, where the U shape was preserved (*Appendix 1—figure 4a*).

Next, we incorporated the possibility of asymmetric death rates by introducing a species-specific death rate in addition to a uniform dilution rate:

$$\frac{1}{N_i}\frac{dN_i}{dt} = \sum_{j=1}^{7}\frac{r_{ij}c_j}{K_{ij}+c_j} - \delta - \delta_i \tag{15}$$

The species-dependent death rate $\delta_i$, randomly sampled from a uniform distribution of [-0.1, 0.1], introduced asymmetry to mortality of each species in the community. During simulations, species with high mortality were severely penalized compared to species with low mortality, and the species pool was effectively reduced to only species with low mortality. This modification led to a lower maximum diversity of the system, but U-shaped DDR was maintained (*Appendix 1—figure 4b*).

Finally, we incorporated the possibility of migration by introducing a species-specific migration from a common source:

$$\frac{1}{N_i}\frac{dN_i}{dt} = \sum_{j=1}^{7}\frac{r_{ij}c_j}{K_{ij}+c_j} - \delta + \frac{m_i}{N_i} \tag{16}$$

The migration rate $m_i$, set as 0.001 for all species, is applied continuously regardless of the fluctuation in disturbance, modeling a constant population influx from a highly diverse common source. This uniform migration boosted the diversity overall, yet the U-shaped DDR was preserved (*Appendix 1—figure 4c*).

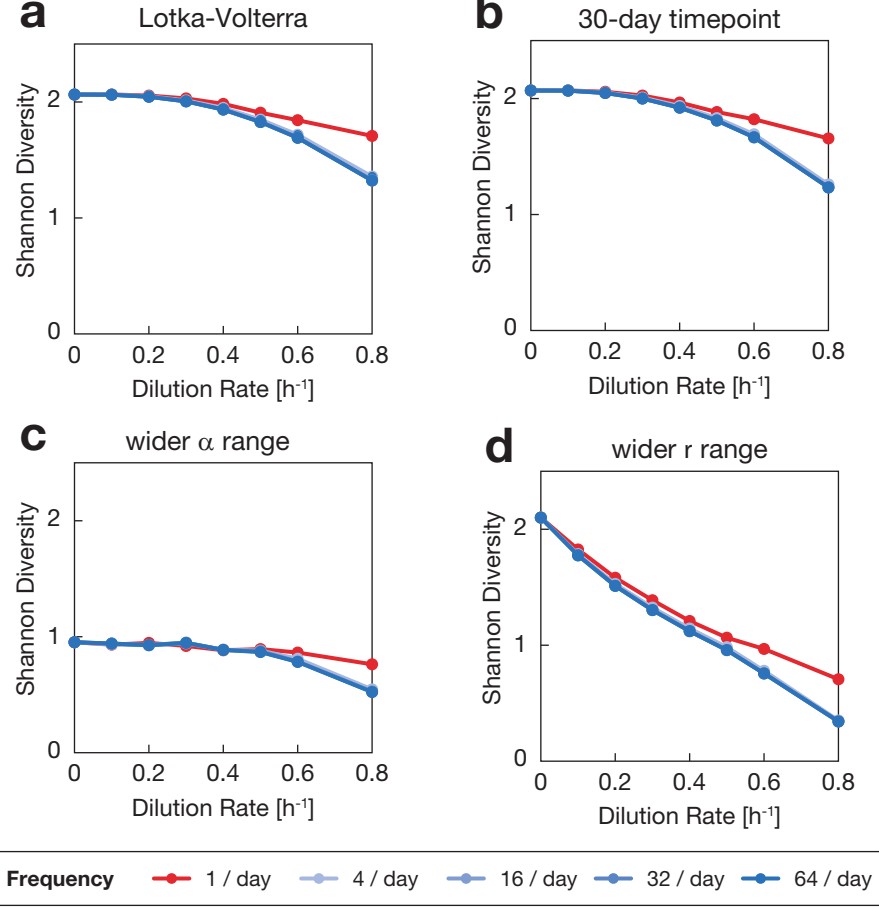

**Appendix 1—figure 1.** Lotka-Volterra models do not exhibit strong relationship between diversity and fluctuation. Shannon diversity was plotted vs. disturbance intensity (mean dilution rate), taking the mean across 100 communities after 6 days of simulated growth. Fluctuations are classified by frequency of dilution disturbance, indicated by color. (**a**) Shannon diversity for Lotka-Volterra model shows little relationship between diversity and fluctuations. (**b**) Longer simulation lengths do not

*Appendix 1—figure 1 continued on next page*

*Appendix 1—figure 1 continued*

change this relationship. (**c**) Simulations were run using $\alpha$ interaction parameters with a wider uniform range ($\alpha$ = 0.5-1.5) and increased mean. The DDR was only minorly affected, retaining a decreasing relationship between diversity and disturbance intensity. (**d**) Simulations were run using growth rate parameters with a wider uniform range ($r$ = 0.1-1) and decreased mean. Though mean diversity dropped, no qualitative change in DDR shape was observed.

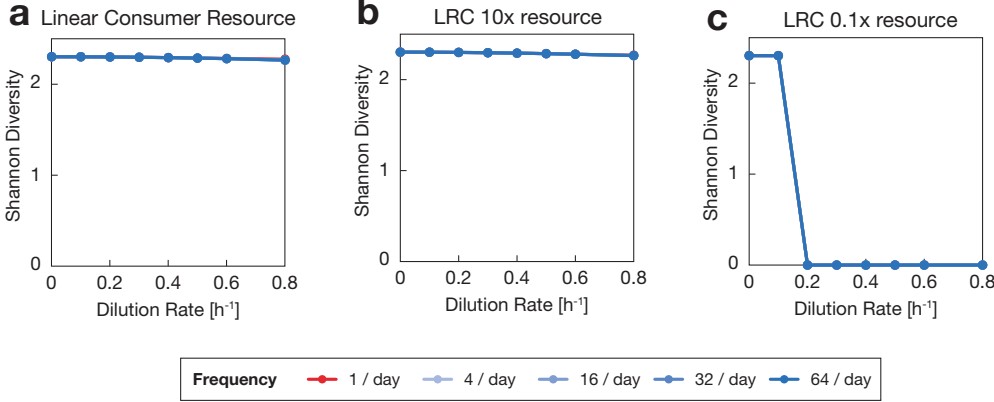

**Appendix 1—figure 2.** Linear Consumer Resource models do not exhibit relationship between diversity and fluctuation. Shannon diversity was plotted vs. disturbance intensity (mean dilution rate), taking the mean across 100 communities after 6 days of simulated growth. Fluctuations are classified by frequency of dilution disturbance, indicated by color. (**a**) Shannon diversity for linear consumer resource (LRC) model shows no relationship between diversity and fluctuations. (**b**) Simulations were run with increased resource supply concentrations ($10 \times c_{jo}$), which did not affect the DDR. (**c**) Simulations were run with decreased resource supply concentrations ($0.1 \times c_{jo}$). This reduced growth rates to a level below the dilution rate, leading to washout and a corresponding Shannon diversity of zero.

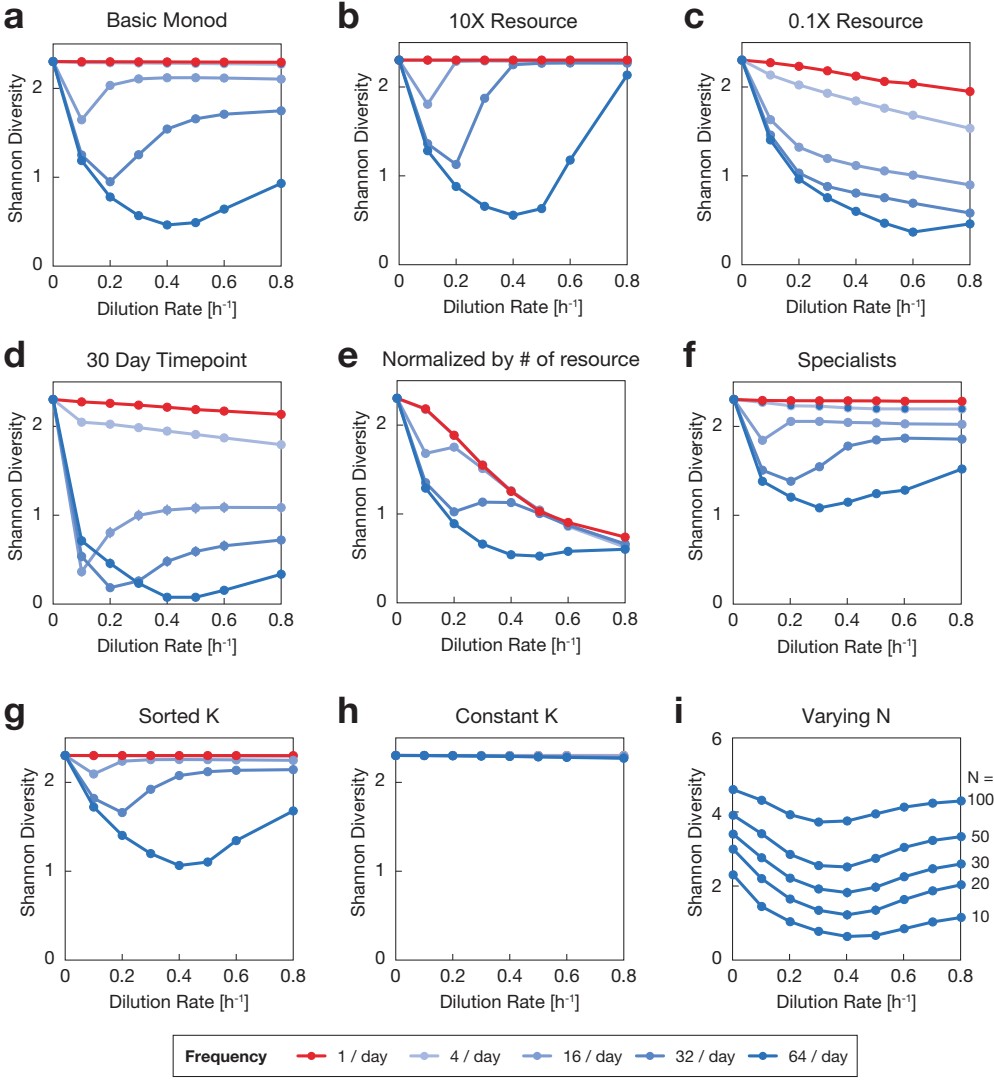

**Appendix 1—figure 3.** Comparison between Monod consumer resource models with variant parameter choices reveal those that retain U-shape. Each plot displays Shannon diversity after 6 days of simulated growth, mean across 100 communities for varying mean dilution rate and dilution frequency (see legend). Variations on the basic Monod model are discussed in Materials and methods. (**a**) Basic Monod Growth model, reproduced from *Figure 3e*. (**b**) Increasing resource supply by a factor of 10 does not qualitatively change the DDR. (**c**) Decreasing resource supply by a factor of 10 strongly affects the DDR by moving the competition out of the saturating nutrient range (set by $K$), such that the Monod model approaches a linear consumer resource model. (**d**) Carrying out the basic Monod growth model from a over longer simulation lengths retains the qualitative DDR shape. (**e**) Randomized growth rates on each resource were normalized by the number of resources rather than the species summed growth rates, which permitted wide variance in overall species growth rates, but retained some U-shape. (**f**) Simulations of communities of 10 species of specialists that consumed only 3 out of 7 resources exhibited higher diversity, but retained U-shaped DDRs. (**g**) Sorted $K$ simulations, reproduced from *Appendix 2—figure 1*. (**h**) Constant $K$ simulations, in which a randomly selected $K$ is shared for all species and resources, do not show U-shape. (**i**) Simulations of communities with varying N number of species and equal number of N resources exhibit U-shaped DDRs under 64/day dilution frequencies.

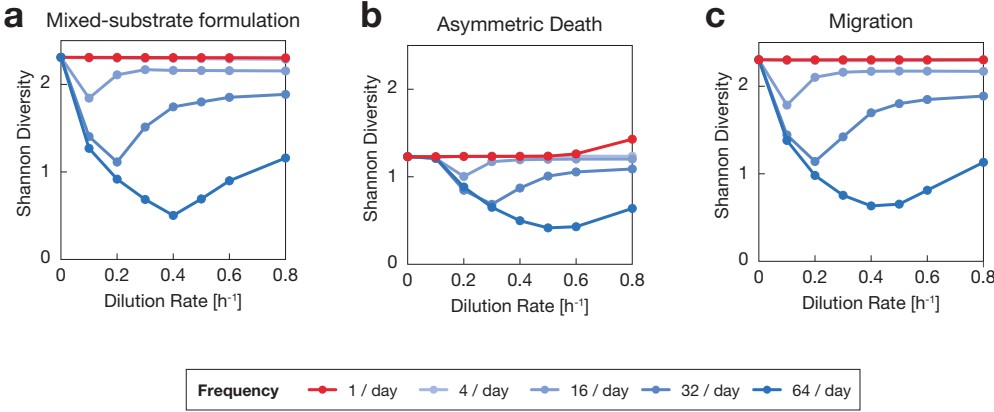

**Appendix 1—figure 4.** Comparison between Monod consumer resource models with variant assumptions can retain U-shape. Each plot displays Shannon diversity after 6 days of simulated growth, mean across 100 communities for varying mean dilution rate and dilution frequency (see legend). Variations on the basic Monod model are discussed in Materials and methods. (**a**) Growth rates across multiple resources were normalized using an alternative mixed-substrate utilization model, retaining the DDR shapes. (**b**) Rather than having equal mortality rates across species, each species was assigned an independent mortality rate $\delta_i$ in addition to the universal dilution rate $\delta$. Diversity is reduced as species with higher mortality rates are penalized, but this does not qualitatively change the DDRs observed. (**c**) To account for migration, each species was assigned a migration rate from a universal source community, independent of the dilution rate. Though species are constantly reintroduced into the community, if the migration rates are low compared to growth and dilution rates, then there are no major qualitative changes to the observed DDRs.

## Appendix 2

### Theoretical analysis of the niche-flip mechanism underlying U-shaped diversity disturbance relationships

As discussed in the main text, the U-shaped diversity disturbance relationship we observed under constantly applied disturbance has not been extensively explored in the literature. In a theoretical study based on this observation, we describe how a mechanism we term 'niche-flip' can lead to a U-shaped diversity-disturbance relationship. To introduce niche flip, we focus on a two-species, two-resource community. In this supplementary section, we will mathematically show how $r/K$ tradeoffs can lead to niche flip. Without losing generality, these conditions can be summarized qualitatively:

- At low disturbance intensities, species 1 drives species 2 extinct in resource 1 and species 2 drives species 1 extinct in resource 2. Species 1 and 2 can coexist in the presence of both resource 1 and 2.
- At high disturbance intensities, species 2 drives species 1 extinct in resource 1 and species 1 drives species 2 extinct in resource 2. Species 1 and 2 can coexist in the presence of both resource 1 and 2.

We assume that the growth rate of each species is represented by a summed Monod model:

$$\frac{1}{N_i}\frac{dN_i}{dt} = \sum_j r_{ij}\frac{c_j}{K_{ij}+c_j} - \delta \tag{17}$$

$N_i$ is the population of species $i$, $c_j$ is the concentration of resource $j$, $r_{ij}$ is the maximum growth rate of species $i$ on resource $j$, $K_{ij}$ is the Monod constant of species $i$ on resource $j$, and $\delta$ is a global death rate added to the system, such as the dilution rate in our experiments. The magnitude of $\delta$ is the disturbance intensity in the system.

We consider chemostat-like continuous dynamics, where resources are supplied at the same rate at which they are diluted, and they are consumed due to the species' growth:

$$\frac{dc_j}{dt} = -\sum_i a_{ij}r_{ij}\frac{c_j}{K_{ij}+c_j} + \delta\left(c_{j0}-c_j\right) \tag{18}$$

$a_{ij}$ is the inverse yield of resource $j$ for species $i$ (i.e. how much of resource $j$ is needed for a unit growth in population $i$), and $c_{j0}$ is the fresh media concentration of supplied resource $j$. For simplicity, we will assume $a_{ij}=1$ throughout our analysis.

We note that the competition result under a single supplied resource is determined by the equilibrium resource concentration $c_{ij}^*$ when species $i$ survives entirely on resource $j$. $c_{ij}^*$ is then a solution of:

$$r_{ij}\frac{c_{ij}^*}{K_{ij}+c_{ij}^*} = \delta \tag{19}$$

Then species 1 drives species 2 extinct in resource 1 if and only if:

$$c_{11}^* < c_{21}^* \tag{20}$$

Let us consider two disturbance regimes separately, and describe the necessary conditions for coexistence in the presence of both resources in each case:

### Condition 1: Low dilution regime: $\delta \ll r_{ij}$

In this case, the resource concentrations at equilibrium are much smaller than the Monod constant: $c_j^* \ll K_{ij}$. Under this regime, we can approximate the growth rate to be simply proportional to resource concentration:

$$\frac{1}{N_i}\frac{dN_i}{dt} = \sum_j r_{ij}\frac{c_j}{K_{ij}+c_j} - \delta \simeq \sum_j \frac{r_{ij}}{K_{ij}}c_j - \delta \tag{21}$$

Here, the equilibrium resource concentration becomes:

$$c_{ij}^* = \delta \frac{K_{ij}}{r_{ij}} \tag{22}$$

The condition for species 1 to win in resource 1 while species 2 wins in resource 2 is then:

$$\frac{K_{11}}{r_{11}} < \frac{K_{21}}{r_{21}}, \frac{K_{12}}{r_{12}} > \frac{K_{22}}{r_{22}} \tag{23}$$

## Condition 2: High dilution regime: $\delta < r_{ij}, \delta \sim r_{ij}$

In this case, $\delta$ is large and approaches (but does not exceed) the maximum growth rate for each species on each resource. This limit is necessary in order to obtain positive solutions for $c_{ij}^*$:

$$c_{ij}^* = \frac{\delta K_{ij}}{r_{ij} - \delta} \tag{24}$$

Then the condition for niche flip becomes:

$$\frac{K_{11}}{r_{11} - \delta} > \frac{K_{21}}{r_{21} - \delta}, \frac{K_{12}}{r_{12} - \delta} < \frac{K_{22}}{r_{22} - \delta} \tag{25}$$

Focusing on resource 1, combining the condition for low- and high-disturbance leads to:

$$\frac{r_{11}}{r_{21}} > \frac{K_{11}}{K_{21}} > \frac{r_{11} - \delta}{r_{21} - \delta} \tag{26}$$

Under the assumption that $r_{ij} > \delta$, these inequalities are equivalent to:

$$r_{21} > r_{11}, K_{21} > K_{11} \frac{r_{21}}{r_{11}} \tag{27}$$

Recall that in the high-dilution regime, species 2 wins in resource 1 and species 1 wins in resource 2. Similarly, for resource 2, we have another inequality in the reverse direction:

$$r_{22} < r_{12}, K_{22} < K_{12} \frac{r_{22}}{r_{12}} \tag{28}$$

Thus, for dilution regimes where each resource is capable of supporting each species' survival, two requirements must be met in order to enable niche flip. First, at a given dilution rate, each species should win on a different resource. Second, the outcome of competition on each resource should flip between low and high dilution regime, which is possible if r/K tradeoffs occur on each resource.

Additionally, based on these results, we can build intuition about multi-species, multi-resource communities, similar to those in our experiments. In order to exclude all instances of niche flip that would lead to a U-shaped DDR in a multi-species community in complex media, every pair of species must not exhibit r/K tradeoffs in all pairs of resources. This implies that in order for the community to not exhibit a U-shaped DDR, the tradeoff condition (*Equation 25* above) should not be met among all pairs in all resources, and instead the following reverse condition should be satisfied for all $i, j, k$:

$$r_{ij} < r_{kj} \, iff \, \frac{r_{ij}}{K_{ij}} < \frac{r_{kj}}{K_{kj}} \tag{29}$$

Furthermore, we can extend this condition to overall parameters measured in a complex media, rather than on each component resource. Summed Monod growth on all resources can be approximated by a single Monod growth curve with:

$$r_i = \sum_j r_{ij}, K_i = \frac{r_i}{\sum_j \frac{r_{ij}}{K_{ij}}} \tag{30}$$

To satisfy the condition for avoiding a U-shaped DDR, the community should show a positive correlation between $r_i$ and $r_i/K_i$:

$$r_i < r_k \; iff \; \frac{r_i}{K_i} < \frac{r_k}{K_k} \qquad (31)$$

In conclusion, in the summed Monod model, a U-shaped DDR is guaranteed when $r$ and $r/K$ in complex media do not have a positive correlation. To be certain, the strength of the U-shape should depend on the prevalence of tradeoffs, and by extension, a weak correlation would result in a less dramatic U-shape than would a strong negative correlation. Additionally, observing a U-shape resulting from a small number of rare tradeoffs would require measuring the community at the dilution rates where the few instances of niche flip occur. The lack of positive correlation exhibited by our isolates (*Figure 3—figure supplement 4*), however, indicates that niche flip is a relevant mechanism that warrants further investigation.

Thus far, we derived the required conditions for niche flip in a two-species, two-resource community:

$$r_{21} > r_{11}, K_{21} > K_{11}\frac{r_{21}}{r_{11}}, r_{22} < r_{12}, K_{22} < K_{12}\frac{r_{22}}{r_{12}} \qquad (32)$$

These inequalities represent trade-offs between $r$ and $K$ that cause the Monod growth curves of the two species in both resources to cross, but such that the species that grows faster at low resource concentration flips between the two resources (*Figure 2—figure supplement 1*).

As discussed in the main text, these $r/K$ tradeoffs make coexistence possible at low and high disturbances, while at some level of intermediate disturbance the niche flip eliminates the possibility for coexistence. This mechanism could lead to a U-shaped diversity-disturbance relationship among a community of many species consuming many resources, assuming that trade-offs between $r$ and $K$ were prevalent. In the following pages, we show that this requirement can be overcome, as $r/K$ trade-offs across different resources can also lead to niche flip. To illustrate this concept, we must consider an even higher disturbance regime than explored previously.

Let us first check whether eliminating $r/K$ tradeoffs on each resource would eliminate the U-shaped DDR. In a simulation of a 10-species, 7-resource community, we sorted $r$ and $K$ such that the Monod growth curves on each of the seven resources do no intersect, in order to avoid trade-offs in each of the seven resources (*Appendix 2—figure 1a*). Surprisingly, this sorting did not eliminate the overall U-shaped DDR (*Appendix 2—figure 1b*). To explain how a lack of $r/K$ tradeoffs on each resource could nevertheless lead to niche flip and the U shape, we return to the example of a two-species, two-resource community. Let us consider an even higher disturbance regime than explored previously.

## Condition 3: Very high dilution regime: $\delta > r_{ij}$

In this regime, the dilution rate is greater than any species' maximum growth rate on any single resource. In the summed Monod model, species can survive when the sum of growth rates across resources can overcome the dilution rate.

In the two-species, two-resource example, neither species can survive on a single resource, but both could survive in the presence of both resources at a sufficient concentration. The niche of each species is now determined not by the intersection of the ZNGIs and the resource axis, but by the asymptotic distance between the ZNGI and the resource axis. A species that requires less additional resource 2 in order to survive would win in media containing mostly resource 1. For this reason, growth on a second resource now plays a role in determining the winner in the (predominantly) first resource. Under the summed Monod model, the necessary concentration of resource 2 can be solved explicitly:

$$r_{i1} + r_{i2}\frac{c_{i2}^*}{K_{i2} + c_{i2}^*} = \delta \qquad (33)$$

Solving this gives:

$$c_{i2}^* = K_{i2} \frac{\delta - r_{i1}}{r_{i1} + r_{i2} - \delta} \tag{34}$$

As a reminder, in the low dilution limit, the competition outcome on resource 1 is determined by:

$$c_{i1}^* = K_{i1} \frac{\delta}{r_{i1}} \tag{35}$$

Without losing generality, let us assume species 1 outcompetes species 2 at low dilution rates in resource 1. Then the niche flip requires that the outcome flips as we change the dilution rate and species 2 outcompetes species 1 in resource 1 under the high-dilution regime. Then the niche flip condition becomes:

$$\frac{K_{11}}{r_{11}} < \frac{K_{21}}{r_{21}}, K_{12} \frac{\delta - r_{11}}{r_{11} + r_{12} - \delta} > K_{22} \frac{\delta - r_{21}}{r_{21} + r_{22} - \delta} \tag{36}$$

Now the competition outcome under large disturbance depends not only on the maximum growth rate on the particular resource but also on the Monod constant on the other resource, which determines the extra amount of the other resource necessary to overcome the dilution rate. This coupling between two resources opens up additional conditions for niche flip without $r/K$ trade-offs in each resource. For example, under simplifying assumption that $r_{11} + r_{12} = r_{21} + r_{22}$, one can have niche flip with

$$r_{11} > r_{21}, K_{12} < \frac{\delta - r_{12}}{\delta - r_{22}} K_{22}, r_{12} < r_{22}, K_{11} > \frac{\delta - r_{11}}{\delta - r_{21}} K_{21} \tag{37}$$

In this example, there are no apparent $r/K$ trade-offs on each resource. The trade-off between $r_{i1}$ and $K_{i2}$ leads to the flipped outcome on resource 1, and similarly the trade-off between $r_{i2}$ and $K_{i1}$ leads to the flipped outcome on resource 2. The niche-flip may be observed on a Tilman diagram as before (*Appendix 2—figure 1*).

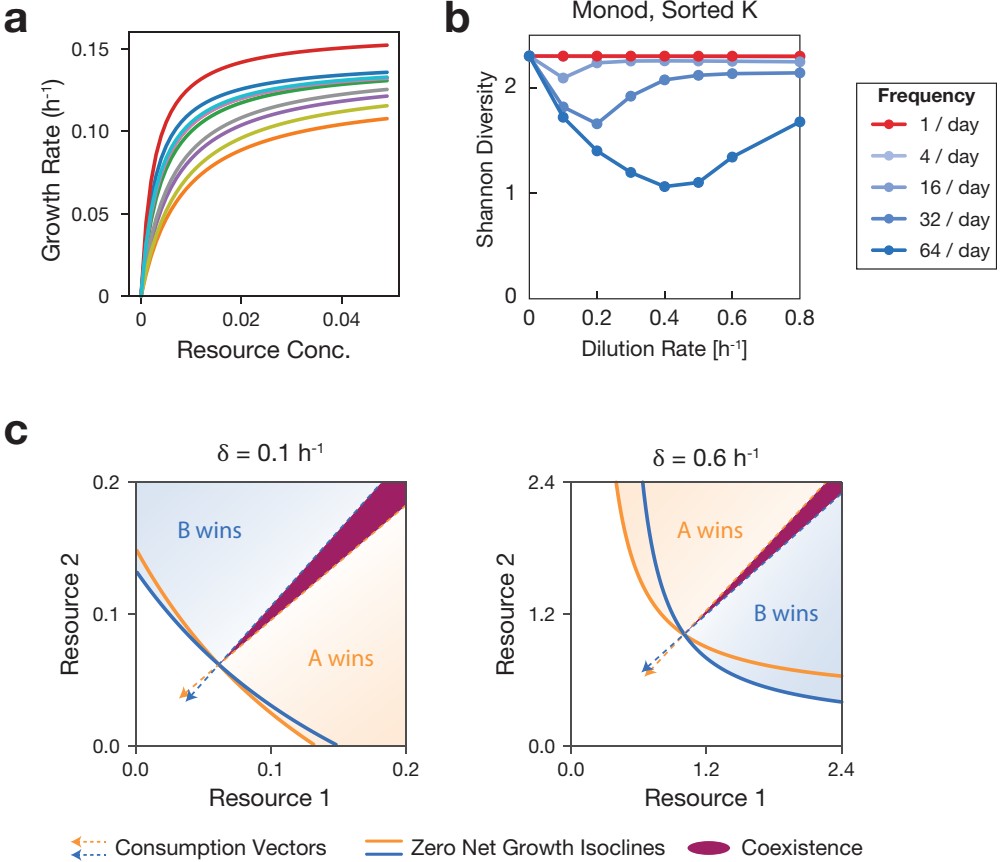

**Appendix 2—figure 1.** Generating species with sorted *r* and *K* parameters prevents intersection of growth curves but not niche flip. (**a**) Growth curves of 10 species on a single resource are shown. For these species in a single resource environment, the outcome of competition does not depend on resource concentration or dilution rate. (**b**) Communities with sorted *K* per resource still exhibit U-shaped DDR. Shannon diversity of communities vs. mean dilution rate. Color indicates frequency of fluctuations. Calculations performed after 6 days of simulated growth, and averaged across 100 communities with randomly generated and sorted *r/K* parameters. (**c**) Niche flip occurs in a community with sorted *K* parameters at high dilution rate. At low dilution rate, ZNGIs intersect the axes and the resource consumption vectors outline a coexistence region. At high dilution rate, species cannot survive on a single resource, so ZNGIs do not intersect the axis. The relative positions of ZNGIs and consumption vectors (i.e. invasion boundaries) have flipped at high dilution rate, consistent with niche-flip.

Therefore, since *r/K* trade-offs on separate resources can also lead to niche flip, we must keep *K* constant for all species grown on any particular resource in order to eliminate the possibility of niche flip. In a simulation of a ten-species, seven-resource community, with one *K* per resource assigned to all ten species, the diversity-disturbance relationship is not U-shaped (*Appendix 1—figure 3*).

In conclusion, we observe that when the dilution rate is smaller than the maximal growth rate on each resource, *r/K* tradeoffs on each resource are required for niche-flip and a U-shaped diversity-disturbance relationship. If we consider an even higher dilution rate, *r/K* tradeoffs on each resource are no longer required for niche-flip because *r/K* tradeoffs across different resources can also lead to niche-flip. This opens up additional parameter space that would lead to a U-shape diversity-disturbance relationship.

In a multi-resource multi-species community, this implies that even a positive correlation between *r* and *r/K* in complex media does not necessarily guarantee the absence of a U-shaped DDR. Our results show that niche flip and a resulting U-shaped DDR under constant disturbance are difficult to avoid in the Monod model, and that their absence requires a very constrained parameter space. Note however, that in this section we have considered only chemostat continuous dynamics. As

shown experimentally and in simulations, introducing fluctuations into the system entirely reshapes the DDRs (*Figure 3e and f*). Finally, while the simulations are robust to a variety of parameter choices and alternative formulations (*Appendix 1—figure 3* and *4*), populating the model with growth parameters from different species-media combinations may yield a diverse range of DDR shapes. Nevertheless, we believe the interesting features of the niche-flip mechanism could warrant its study in other ecosystems.

