## [Decision Letter]

**Acceptance summary:**

How species diversity responds to external perturbations (such as resource influxes and dilutions) is an important ecological question. Using soil microbial communities in devices where perturbations can be introduced in various forms and mathematical modelling, this study nicely illustrates how diversity is influenced by perturbations.

**Decision letter after peer review:**

Thank you for submitting your article "Environmental fluctuations reshape an unexpected diversity-disturbance relationship in a microbial community" for consideration by *eLife*. Your article has been reviewed by 3 peer reviewers, including Wenying Shou as the Reviewing Editor and Reviewer #1, and the evaluation has been overseen by Aleksandra Walczak as the Senior Editor.

All three reviewers liked the topic and the combination of theory and experiments.

Essential revisions include:

1) Bridge your work with the literature on disturbance ecology literature.

2) Address the point that mortality is both a disturbance event and an outcome in the continuous culture system.

3) Application to systems other than continuous cultures.

4) Analyze compositional data.

5) Address the relevance of tradeoff, especially in more complex communities.

*Reviewer #1 (Recommendations for the authors):*

During the presubmission stage, I have already given authors suggestions on how to improve their writing. They have greatly improved the clarity of their writing. As a result, the current version is a pleasure to read. The work is thorough, and nicely combines experiments and modelling.

I have no major critiques. I have one comment: Presumably, when dilution rate increases more, species diversity will drop again as the dilution rate exceeds the maximal growth rates of many species? Authors should perhaps discuss that? Authors did discuss biofilm formation and how species persisted despite high dilution rate. I find this interesting.

Figure 2. I am fairly sure that panel a is mislabeled. Low Resource 1 and high Resource 2: A should win according to the left panels.

*Reviewer #2 (Recommendations for the authors):*

Thank you for the opportunity to review this interesting work.

Question of application to non-equilibrium systems

Would like to see discussion of how the author's implementation of DDR would apply (or not) to non-equilibrium ecosystems. Or, is this outcome most useful in a context similar to continuous culture? Related, ecosystem disturbances often alter not only the amount of resources available (high/low concentrations), but also the types or number of resources available, potentially shifting competitive advantages to entirely different sets of taxa. A next extension could be to account for those resource dynamics in the modeling.

A philosophical note: One may perceive that the condition of "constant" disturbance/mortality with continuous flow is not a disturbance to the continuous culture system, it is the equilibrium condition, analogous to a lentic aquatic system. The flow-through is the "normal", the disturbance would be the stoppage of the flow or periodic interruption of it, perhaps in the lentic system as during a mid-summer drought. With this idea, there would be some situations in which the "constant" change in the environment is actually the low disturbance intensity, and the periodic change in the environment is intermediate, and abrupt end of environmental change high disturbance. Again, the disturbance is defined relative to the ecosystem of interest – would be interesting to discuss or at mention how the ecosystem traits may result in the inverse disturbance intensity gradient from what is used here, and how that may impact the outcome/interpretation

Though I understand the importance of simple models for developing quantitative theory, I wonder about the application of this particular U-shaped DDR and modeling approach to more realistic communities that contain more than ~10 taxa. At minimum, suggest to discuss the implications of the DDR and modeling given more diverse communities – with hundreds to thousands of taxa? Does it still hold?

*Reviewer #3 (Recommendations for the authors):*

Here are some major comments and suggestions to the authors.

1) Diversity in data and theory? The data clearly shows that the U-shape of the diversity-disturbance relationship (DDR) is a robust and reproducible feature of experiments. Moreover, Figure S10 clearly shows that different measures of diversity (richness and Shannon diversity) follow the same patterns for the data. Is this true also for the model?

2) Parameter choice. The authors considered 10 species competing for 7 resources. This choice can support at stationarity at most 7 species. Figure S10 shows that, at least for some conditions, many more "species" coexist. Is the particular choice of the number of species in the initial pool and the number of resources important? Does it affect the final results?

3) Importance of the tradeoff (in theory). Figure 2 emphasizes the importance of the tradeoff to see the niche flip. This is also discussed explicitly in the supplement in the two species and two resources case. There it is shown that, for a very high dilution rate, the tradeoff "across different resources" can also lead to niche-flip (which is shown in Figure 16c). It is unclear to me, in the end, if the tradeoff matters or not for the niche flip. If it does not matter, what is the essential ingredient? I think that this aspect should be better discussed in the text.

4) "If we consider an even higher dilution rate, r/K tradeoffs on each resource are no longer required for niche-flip because r/K tradeoffs across different resources can also lead to niche-flip." This argument is clear for two species / two resources. But I do not understand what r/K tradeoff across different resources means for more than two resources. Does it mean that it is enough to find a pair of resources such that such a tradeoff exists?

5) Existence of the tradeoff (in data). Figure S7 cannot be used as evidence of the tradeoff. The existence of an "unoccupied parameter space" in the top right of the plot is driven only by the presence of one data point with a significantly higher value of r/K. In other words, I do not think that the correlation between r/K and r is statistically significant.

---

## [Author Response]

Reviewer #1 (Recommendations for the authors):During the presubmission stage, I have already given authors suggestions on how to improve their writing. They have greatly improved the clarity of their writing. As a result, the current version is a pleasure to read. The work is thorough, and nicely combines experiments and modelling.

Thank you for providing those excellent suggestions. We agree that they have significantly improved the clarity of the writing.

I have no major critiques. I have one comment: Presumably, when dilution rate increases more, species diversity will drop again as the dilution rate exceeds the maximal growth rates of many species? Authors should perhaps discuss that? Authors did discuss biofilm formation and how species persisted despite high dilution rate. I find this interesting.

This is an excellent point. As predicted by simulations (Figure 4), diversity would be expected to decrease as the dilution rate exceeds the maximal growth rates of species in the community. This was the motivation behind running the washout experiment, with dilution rates of up to 1.5h^-1^; however, as we mentioned in the text, we were not able to see washout in our experimental system (Figure 3 —figure supplement 3). As you noted, we believe that biofilm and/or incomplete mixing at extreme dilution rates (particularly in the 1/day disturbance events) could be main contributing factors. We did observe some biofilm formation (Figure 3 —figure supplement 12), so this is our working hypothesis for why we fail to see loss of diversity in our washout conditions.

Figure 2. I am fairly sure that panel a is mislabeled. Low Resource 1 and high Resource 2: A should win according to the left panels.

The labeling is correct; we apologize for any confusion. During the pre-submission stage, we changed the labeling in order to distinguish between the growth rate (Species A or B faster) and the outcome of competition (Species A or B wins). As shown in more detail in Figure 2 —figure supplement 1, since cells consume resources, the resource levels decrease until the growth rate is matched by the dilution rate. Therefore, the species that can grow at a rate equal to the dilution rate with the fewest resources (lowest c*) will be the winner. At low dilution rates, such as those plotted in Figure 2A, equilibrium resource levels will be low; in this case, when growing on Resource 2, Species B will win the competition because it has a lower c* and is therefore more efficient and can survive at lower resource levels. We have added the following line to the figure caption.

“Resources are consumed until reaching equilibrium along the ZNGI of the species requiring the fewest resources to survive at the specified mortality rate, the winner of the competition.”

We have also included the relevant text from the main (below), but we are open to any suggestions aimed at making this point more clear:

“Competitive outcomes at low disturbance intensities will depend on the relative growth rates at low resource concentrations, while competitive outcomes at high disturbance intensities will depend on the relative growth rates at high concentrations.”

Reviewer #2 (Recommendations for the authors):Thank you for the opportunity to review this interesting work.Question of application to non-equilibrium systemsWould like to see discussion of how the author's implementation of DDR would apply (or not) to non-equilibrium ecosystems. Or, is this outcome most useful in a context similar to continuous culture? Related, ecosystem disturbances often alter not only the amount of resources available (high/low concentrations), but also the types or number of resources available, potentially shifting competitive advantages to entirely different sets of taxa. A next extension could be to account for those resource dynamics in the modeling.

Thank you for this very interesting suggestion. Indeed, systems far from either equilibrium or oscillatory disturbance patterns may lead to more chaotic changes in disturbance outcomes. While niche flips are likely to arise from new resource supply points, the relative importance of these effects in the face of changing resource supplies is likely smaller than in systems that have opportunities to equilibrate. While these systems could by tackled by our simulations, our focus in this work is on systems that resemble our experimental system and to highlight the amount of complexity that can arise from a simple system without invoking more complex interventions. We have added the following note to the discussion to highlight this point:

“Intriguingly, these complex ecosystem assembly rules can emerge from temporal structure alone, without invoking other organizing principles, such as spatial structure^44^ or network structure (e.g. cross-feeding and antagonism)^45,46^. While we considered both constant and oscillating temporal patterns of disturbance, additional complexity may arise from temporal patterns that are random or those that change which types of resources available over time. Since complexity can arise from temporal, spatial, and network effects, the relative importance of any one organizing factor is likely ecosystem dependent.”

A philosophical note: One may perceive that the condition of "constant" disturbance/mortality with continuous flow is not a disturbance to the continuous culture system, it is the equilibrium condition, analogous to a lentic aquatic system. The flow-through is the "normal", the disturbance would be the stoppage of the flow or periodic interruption of it, perhaps in the lentic system as during a mid-summer drought. With this idea, there would be some situations in which the "constant" change in the environment is actually the low disturbance intensity, and the periodic change in the environment is intermediate, and abrupt end of environmental change high disturbance. Again, the disturbance is defined relative to the ecosystem of interest – would be interesting to discuss or at mention how the ecosystem traits may result in the inverse disturbance intensity gradient from what is used here, and how that may impact the outcome/interpretation

Thank you for this interesting suggestion. We agree that our “constant” disturbance condition is reflective of the baseline state of lentic aquatic systems. Similarly, our low frequency disturbance conditions could be considered systems with high flow rates and long periods of drought. A further exploration could include 4 parameters: maximum flow, minimum flow, duration at maximum and duration at minimum. We have added a paragraph in the Discussion section to address this potential extension which would also enable us to explicitly and independently test parameters defined in other studies (like Miller et al.,):

“While we focused on two characteristics, a number of different temporal features of a disturbance could be considered^7,17,18^. In our present work, by focusing on mean disturbance intensity rather than maximum disturbance intensity, we have sampled conditions with a negative correlation between frequency and maximum intensity (e.g. comparing frequent small dilution events to infrequent large dilution events). Furthermore, our definition of constant disturbance may in fact represent the undisturbed state of some ecosystems. For natural ecosystems with constant flow, such as lentic aquatic systems, it may be more accurate to consider a drought disturbance with a sudden reduction in flow rate (with distinct biological outcomes). Performing an experiment in which the maximum flow rate, minimum flow rate, duration at maximum and duration at minimum were all independently varied could help to understand a wider range of disturbance profiles.”

Though I understand the importance of simple models for developing quantitative theory, I wonder about the application of this particular U-shaped DDR and modeling approach to more realistic communities that contain more than ~10 taxa. At minimum, suggest to discuss the implications of the DDR and modeling given more diverse communities – with hundreds to thousands of taxa? Does it still hold?

We appreciate this suggestion. In principle the niche-flip mechanism that leads to the U-shape should hold in more diverse communities as well. This is because niche-flip of a pair of species under a pair of resources is sufficient to prevent the pair’s coexistence in the multi-resource environment, reducing the total diversity. To test this idea, we performed simulations of up to 100 species with a proportional number of resources and obtained the U-shaped DDR under constant disturbance conditions. The U-shape is retained at higher number of species, though its relative impact on the total diversity decreases for more diverse communities. We have revised the manuscript by adding a panel to Appendix figure 3 and referring to it in the main text:

“At increased number of species and resources, a U-shaped DDR is still observed, though its relative impact on total diversity is reduced (Appendix figure 3i).”

Reviewer #3 (Recommendations for the authors):Here are some major comments and suggestions to the authors.1) Diversity in data and theory? The data clearly shows that the U-shape of the diversity-disturbance relationship (DDR) is a robust and reproducible feature of experiments. Moreover, Figure S10 clearly shows that different measures of diversity (richness and Shannon diversity) follow the same patterns for the data. Is this true also for the model?

Thank you for pointing this out. Indeed, we show that different metrics of diversity and DDR from our experiment are robust to this choice. This robustness holds for the model as well, with the caveat that richness thresholds are arbitrary. In particular, the U-shaped curve under high dilution frequency persists when we use richness instead of Shannon diversity as a diversity measure.

To clarify this point, we added a Figure 3 —figure supplement 5 which reproduces the DDR curve from simulations using richness at two different population size thresholds.

2) Parameter choice. The authors considered 10 species competing for 7 resources. This choice can support at stationarity at most 7 species. Figure S10 shows that, at least for some conditions, many more "species" coexist. Is the particular choice of the number of species in the initial pool and the number of resources important? Does it affect the final results?

Thank you for pointing this out. While we observed many more species than 7-10 in our experiment, we chose 10 species and 7 resources to demonstrate a sufficiently complex community with low computational cost. This arbitrary choice is not important in demonstrating the U-shape, as suggested by 2 species – 2 resource case. To show this more explicitly, we performed simulations of up to 100 species with a proportional number of resources. This has been added as a panel to Appendix figure 3, see reference in response to reviewer 2.

3) Importance of the tradeoff (in theory). Figure 2 emphasizes the importance of the tradeoff to see the niche flip. This is also discussed explicitly in the supplement in the two species and two resources case. There it is shown that, for a very high dilution rate, the tradeoff "across different resources" can also lead to niche-flip (which is shown in Figure 16c). It is unclear to me, in the end, if the tradeoff matters or not for the niche flip. If it does not matter, what is the essential ingredient? I think that this aspect should be better discussed in the text.

Thank you for addressing this important point. The essential ingredient for the U-shaped DDR is the niche-flip mechanism, but niche-flip does not require the simple type of tradeoff shown in Figure 2. We present this tradeoff in parameters in the case of two species and two resources as a simple pedagogical example of niche-flip, but as you pointed out, this is not the necessary condition for niche-flip in general.

First, for the extreme dilution case, the geometric analysis in Appendix 2 for condition 3 shows that the conditions for niche-flip are no longer the simple tradeoff. This is because the ZNGIs no longer meet with the axes (i.e. single resource cannot support zero net growth), and the condition for winning in resource 1-dominated environment requires better growth on resource 2 and vice versa. This is manifest in Appendix figure 5 where niche-flip occurs without crossings in Monod curves. We have also included the relevant part of Appendix 2 below:

*“Condition 3*: Very high dilution regime: δ>rij

In this regime, the dilution rate is greater than any species’ maximum growth rate on any single resource. In the summed Monod model, species can survive when the sum of growth rates across resources can overcome the dilution rate.

In the two-species, two-resource example, neither species can survive on a single resource, but both could survive in the presence of both resources at a sufficient concentration. The niche of each species is now determined not by the intersection of the ZNGIs and the resource axis, but by the asymptotic distance between the ZNGI and the resource axis. A species that requires less additional resource 2 in order to survive would win in media containing mostly resource 1. For this reason, growth on a second resource now plays a role in determining the winner in the (predominantly) first resource.”

Additionally, as we will address in the next comment, a U-shaped DDR requires only that there be at least one pair that exhibits niche flip on at least one pair of resources, so many species in the community may have no such tradeoffs. To summarize, while a simple tradeoff in a two species, two resources case is a useful tool for understanding the niche-flip, it is not a necessary ingredient for either niche-flip or U-shaped DDR more broadly. We applied following changes to the text in order to clarify this notion:

“Finally, we emphasize that the illustrated tradeoff is not the only way to obtain niche-flip and U-shaped DDRs (explored further in Appendix 2). To summarize briefly, in the absence of tradeoffs on any one resource, at sufficiently high dilution rates, niche flip and a U-shaped DDR can arise from combinations of two or more resources such that the winner of competition is different at low vs. high equilibrium resource concentrations.”

4) "If we consider an even higher dilution rate, r/K tradeoffs on each resource are no longer required for niche-flip because r/K tradeoffs across different resources can also lead to niche-flip." This argument is clear for two species / two resources. But I do not understand what r/K tradeoff across different resources means for more than two resources. Does it mean that it is enough to find a pair of resources such that such a tradeoff exists?

Thank you for pointing this out. It is indeed enough to find a pair of resources such that such a tradeoff exists. The reason is simply because niche-flip between two resources is a sufficient condition to have niche-flip and a U-shaped DDR in multi-resource condition. The more niche flip pairs that co-occur, the “deeper” the U-shape becomes.

For the many-species case, a U-shaped DDR only requires niche-flip (either via simple tradeoff or a subtle condition for extreme dilutions) for *any pair* of species. And as the number of species increases, the number of pairs scales in a quadratic manner, and accordingly the chances for having a niche-flip between at least one pair of species increases.

For the many-resource case, the U-shape only requires a niche-flip in *any pair* of resources. This is because the region of coexistence in N-dimensional space of supplied N-resources should vanish when the niche flips between at least one pair of resources so that the region of coexistence vanishes on the specific plane. The loss of the region of coexistence will drive one species to exclude the other and reduce diversity in the community.

In addition, as pointed out, r/K tradeoff across different resources becomes nontrivial for more than two resources. For example, r/K tradeoff in 3-resource case may originate from one of the followings conditions:

1. Simple r/K tradeoff between a pair of resources, as illustrated in the main text.

2. r/K tradeoff across a pair of resources, as explicitly calculated in Appendix 2.

3. r/K tradeoff among all 3 resources, such that the angle between consumption vectors of two species vanishes along a nontrivial plane that is not parallel to any of two-resource planes.

We believe it is out of the scope of this work to explicitly calculate such conditions for niche-flip in a multi-resource environment. Additional possibilities for niche-flip among multiple resources only add to the possibilities for niche-flip among pairs of resources, which only increases the probability of the U shape.

5) Existence of the tradeoff (in data). Figure S7 cannot be used as evidence of the tradeoff. The existence of an "unoccupied parameter space" in the top right of the plot is driven only by the presence of one data point with a significantly higher value of r/K. In other words, I do not think that the correlation between r/K and r is statistically significant.

We appreciate the comment. Indeed, we agree that Figure S7 (now Figure 3 —figure supplement 4) is not an evidence of a simple tradeoff. And similarly, as already addressed for previous comments, a simple tradeoff is not a necessary condition for the U-shape. Rather, any deviation from a perfect positive correlation between r/K and r is sufficient to potentially permit a niche-flip. With this figure, we aim to support the observation of U-shaped DDR by showing that the correlation between r/K and r is not significant, which implies that niche-flip between some pairs may exist and therefore U-shaped DDR should follow. We have removed the “unoccupied parameter space” label from this figure and have amended the caption, reproduced below:

“Scatter plot of Monod r and r/K values indicate that while species vary in Monod r and K, these values are uncorrelated. If there were a strong positive correlation between r and r/K, this would prevent a U-shaped DDR according to the model.”